# A UNIFIED THEORY OF ADAPTIVE STOCHASTIC GRADIENT DESCENT AS BAYESIAN FILTERING

## ABSTRACT

We formulate stochastic gradient descent (SGD) as a novel factorised Bayesian filtering problem, in which each parameter is inferred separately, conditioned on the corresponding backpropagated gradient. Inference in this setting naturally gives rise to BRMSprop and BAdam: Bayesian variants of RMSprop and Adam. Remarkably, the Bayesian approach recovers many features of state-of-the-art adaptive SGD methods, including amongst others root-mean-square normalization, Nesterov acceleration and AdamW. As such, the Bayesian approach provides one explanation for the empirical effectiveness of state-of-the-art adaptive SGD algorithms. Empirically comparing BRMSprop and BAdam with naive RMSprop and Adam on MNIST, we find that Bayesian methods have the potential to considerably reduce test loss and classification error.

## 1 INTRODUCTION

Deep neural networks have recently shown huge success at a range of tasks including machine translation (Wu et al., 2016), dialogue systems (Serban et al., 2016), handwriting generation (Graves, 2013) and image generation (Radford et al., 2015). These successes have been facilitated by the development of a broad range of adaptive SGD methods, including ADAGrad (Duchi et al., 2011), RMSprop (Hinton et al., 2012), Adam (Kingma & Ba, 2015), and variants thereof, including Nesterov acceleratation (Nesterov, 1983; Bengio et al., 2013; Dozat, 2016) and AdamW (Loshchilov & Hutter, 2017). However, such a broad range of approaches raises the question of whether it is possible to obtain a unified theoretical understanding of adaptive SGD methods. Here we provide such a theory by reconciling state-of-the-art adaptive SGD algorithms with very early work that used Bayesian (Kalman) filtering to optimize the parameters of neural networks (Puskorius & Feldkamp, 1991; Sha et al., 1992; Puskorius & Feldkamp, 1994; 2001; Feldkamp et al., 2003; Ollivier, 2017).

There have recently been attempts to connect adaptive SGD algorithms to natural gradient variational inference (VI) (Zhang et al., 2017; Khan et al., 2017; 2018). These approaches give a momentum-free algorithm with a mean-square normalizer, in contrast to perhaps the most popular adaptive method, Adam (Kingma & Ba, 2015), which combines momentum with a root-mean-square normalizer. To achieve a closer match to Adam, they modified their natural gradient VI updates, without a principled justification based on approximate inference, to incorporate momentum (Zhang et al., 2017; Khan et al., 2018), and the root-mean-square normalizer (Khan et al., 2017; 2018). As such, there appears to be only a loose connection between successful adaptive SGD algorithms such as Adam, and natural gradient VI.

There is a formal correspondence between natural gradient VI (Zhang et al., 2017; Khan et al., 2017; 2018) and Bayesian filtering (Ollivier, 2017). While Ollivier (2017) did not examine the relationship between their filtering updates and RMSprop/Adam, the equivalence of this particular filtering approach and natural gradient VI indicates that they would encounter the issues described above, and thus be unable to obtain momentum or the root-mean-square normalizer (Zhang et al., 2017; Khan et al., 2017; 2018). More problematically, Ollivier (2017) introduces dynamics into the Kalman filter, but these dynamics correspond to the "addition of an artificial process noise $Q_t$ proportional to [the posterior covariance] $P_{t-1}$". Thus, their generative model depends on inferences made under that model: a highly unnatural assumption that most likely does not correspond to any "real" generative process.

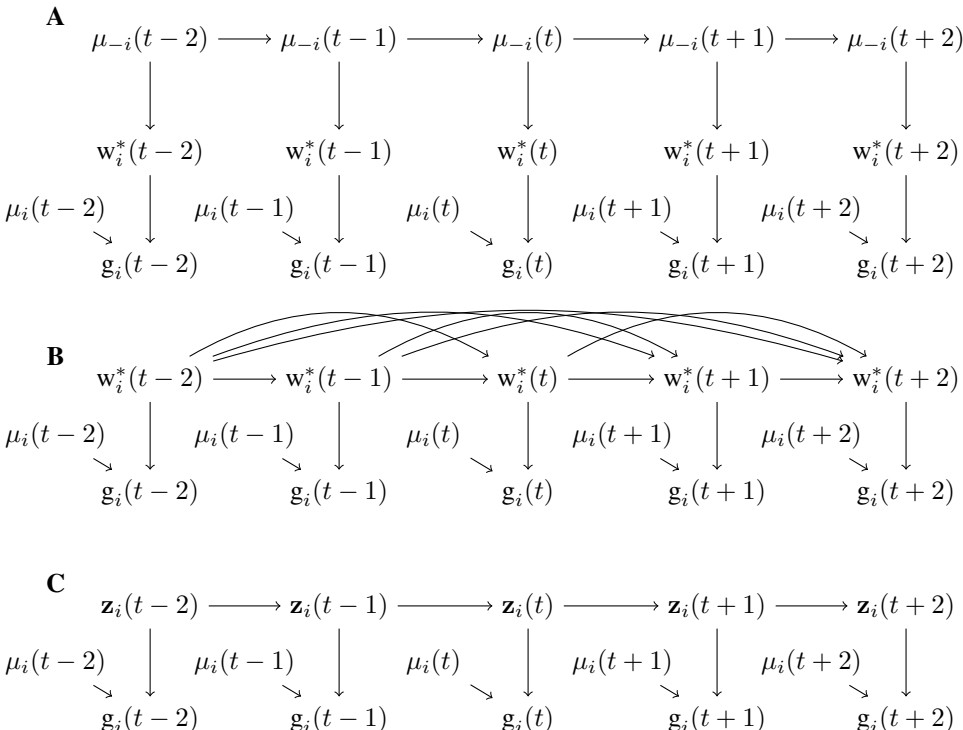

Figure 1: The heirarchy of generative models underlying our updates. **A** Full model for the gradients for a single parameter. The current estimate for all the other parameters, $\mu_{-i}(t)$ vary slowly over time, and give rise to the current optimal value for the $i$th parameter, $\mathbf{w}_i^*$. The gradient then arises from the current estimate of the $i$th parameter, $\mu_i(t)$ (which is treated as an input here), and the optimal value, $i$th parameter, $\mathbf{w}_i^*$. **B** The graphical model obtained by integrating over trajectories for the other parameter estimates, $\mu_{-i}(t)$. In practice, we use a simplified model as reasoning about all possible trajectories of $\mu_{-i}(t)$ is intractable. **C** To convert the model in **B** into a tractable hidden Markov model (HMM), we define a new variable, $\mathbf{z}_i(t)$, which incorporates $\mathbf{w}_i^*$ along with other information about the dynamics.

How might we obtain a principled Bayesian filtering approach that recovers the two key features of state-of-the-art adaptive SGD algorithms: momentum and the root-mean-square normalizer? Here, we note that past approaches including natural gradient VI take a complex generative model over all $N$ parameters jointly, and use a very strong approximation: factorisation. Given that we know that the true posterior is a highly complicated, correlated distribution, it is legitimate to worry that these strong approximations might meaningfully disrupt the ability of Bayesian filtering to give close-to-optimal updates. Here we take an alternative approach, baking factorisation into our generative model, so that we can use Bayesian inference to reason about (Bayes) optimal updates under the constraints imposed by factorisation. In particular, we split up the single large inference problem over all $N$ parameters, $\mathbf{w}$, into $N$ small inference problems over a single parameter. Remarkably, by incorporating factorisation into the problem setting, we convert intractable, high-dimensional correlations in the original posterior into tractable low-dimensional dynamics in the factorised model. This dynamical prior has a "natural" form, at least compared with Ollivier (2017), in that it does not depend on the posterior. Next, we give a generic derivation showing that Bayesian SGD is an adaptive SGD method, where the uncertainty is used to precondition the gradient. We then adapt the generic derivation to the two cases of interest: RMSprop (Hinton et al., 2012) and Adam (Kingma & Ba, 2015). Finally, we discuss the general features of Bayesian adaptive SGD methods, including AdamW (Loshchilov & Hutter, 2017) and Nesterov acceleration (Nesterov, 1983; Dozat, 2016), amongst others.

## 2 FACTORISATION IMPLIES A RICH DYNAMICAL PRIOR

The most obvious factorised approach is to compute the true posterior, $P(w_i|\mathcal{D})$, over a single weight, $w_i$, conditioned on all the data, $\mathcal{D}$. However, this approach immediately fails, because symmetries involving permuting the hidden units (Sussmann, 1992) imply that marginal posteriors such as $P(w_i|\mathcal{D})$ are always so broad as to be useless.

To obtain an informative, narrow posterior, one approach is to eliminate these symmetries by conditioning on our current estimate of the other parameters. In particular, we define a random variable, $w_i^*$, representing the optimal value for the $i$th weight, conditioned on the other weights, $w_{-i}$, being equal to our current estimate of those parameters, $\mu_{-i}$,

$$w_i^* = \arg\max_{w_i} \mathbb{E}\left[\log P\left(d|w_i = w_i, w_{-i} = \mu_{-i}\right)\right],\tag{1}$$

where d is a random minibatch drawn from the true, unknown underlying data distribution (and not just the data we have), so this is analogous to maximum-likelihood with infinite data. As such, even if $\mu_{-i}$ is fixed, $w_i^*$ will be unknown if only have finite data, and therefore we do not know the true underlying data distribution. The optimal weight, $w_i^*$, is a random variable because it depends on our current estimate of the other weights, $\mu_{-i}$, which is a random variable because it depends on a random initialization, and potentially on a randomised optimization algorithm. However, we cannot now use the usual Bayesian approach of inferring $w_i^*$ based on the data, because, in the Bayesian setting, the data are assumed to be generated from a model with parameters, w, not $w_i^*$. Instead, note that the updates for almost all neural network optimization algorithms depend only on the current value of that parameter, and the (history of) backpropagated gradients. This suggests a potential approach: writing down a generative model for the backpropagated gradients that depends on $w_i^*$, then inverting that model to infer $w_i^*$ from those gradients.

Following standard Laplace/Extended Kalman Filter-like approximations (e.g. Zhang et al., 2017; Khan et al., 2017; Ollivier, 2017; Khan et al., 2018) as closely as possible, we approximate the likelihood as a second-order Taylor series expansion. In particular, we consider the log-likelihood for a single minibatch, d, where the minibatch is treated as a random variable drawn from the underlying true data distribution, and we expand in the $i$th direction, keeping the other parameters, $w_{-i}$, fixed to their current estimates, $\mu_{-i}$,

$$\log P\left(d|w_i = w_i, w_{-i} = \mu_{-i}\right) \approx -\tfrac{1}{2}\Lambda_{\text{like},i}\left(w_i - \mu_{\text{like},i}\right)^2 + \text{const},\tag{2}$$

where,

$$\mu_{\text{like},i} = \arg\max_{w_i}\ \log P\left(d|w_i = w_i, w_{-i} = \mu_{-i}\right),\tag{3}$$

$$\Lambda_{\text{like},i} = -\left.\frac{\partial^2 \log P\left(d|w_i = w_i, w_{-i} = \mu_{-i}\right)}{\partial^2 w_i}\right|_{w_i = \mu_{\text{like},i}}.\tag{4}$$

Here, both the mode, $\mu_{\text{like},i}$, and the negative Hessian, $\Lambda_{\text{like},i}$, are random variables as they are deterministic functions of the minibatch, d, and the current estimate of all the other parameters, $\mu_{-i}$, which are both treated as random variables. As such, the gradient of the log-likelihood, which depends on $\Lambda_{\text{like},i}$ and $\mu_{-i}$, is also a random variable,

$$g_i = \left.\frac{\partial \log P\left(d|w_i = w_i, w_{-i} = \mu_{-i}\right)}{\partial w_i}\right|_{w_i = \mu_i} \approx \Lambda_{\text{like},i}\left(\mu_{\text{like},i} - \mu_i\right).\tag{5}$$

where the approximation comes from the second-order Taylor expansion in Eq. (2). Our goal is to understand the distribution of the gradients, conditioned on the random variable $\Lambda_{\text{like},i}$ being equal to some specific value, $\lambda_{\text{like},i}$. In practice, we will set $\Lambda_{\text{like},i}$, using the standard approximations in the natural gradient literature (Zhang et al., 2017; Khan et al., 2017; 2018). The expected gradient is zero at $\mu_i = w_i^*$,

$$0 = \mathbb{E}\left[g_i|w_i^* = w_i^*\right]\Big|_{\mu_i = w_i^*} \approx \mathbb{E}\left[\Lambda_{\text{like},i}\mu_{\text{like},i}|w_i^* = w_i^*\right] - \mathbb{E}\left[\Lambda_{\text{like},i}|w_i^* = w_i^*\right]w_i^*\tag{6}$$

where the approximation again comes only from the second-order Taylor expansion. Thus,

$$\mathbb{E}\left[\mu_{\text{like},i}|\Lambda_{\text{like},i} = \lambda_{\text{like},i}, w_i^* = w_i^*\right] \approx \mathbb{E}\left[\mu_{\text{like},i}|w_i^* = w_i^*\right] \approx w_i^*,\tag{7}$$

with equality if the expected value of $\mu_{\text{like},i}$ is independent of $\Lambda_i$, Substituting this into Eq. (5), we obtain,

$$\mathbb{E}\left[g_i | \Lambda_{\text{like},i} = \lambda_{\text{like},i}, w_i^* = w_i^*\right] \approx \lambda_{\text{like},i}\left(w_i^* - \mu_i\right). \tag{8}$$

However, obtaining an analytic form for the full distribution of the gradient is more difficult. Given that all work in this domain makes fairly strong assumptions (e.g. Zhang et al., 2017; Khan et al., 2017; 2018), an alternative approach opens up: choosing a model for the gradients to match as closely as possible any given approximation of the original likelihood in Eq. (2). In particular, we take,

$$P\left(g_i = g_i | \Lambda_{\text{like},i} = \lambda_{\text{like},i}, w_i^* = w_i^*\right) \approx \mathcal{N}\left(g_i; \lambda_{\text{like},i}\left(w_i^* - \mu_i\right), \lambda_{\text{like},i}\right). \tag{9}$$

Note that the variance is the negative Hessian, $\lambda_{\text{like},i}$, which is reminiscent of Fisher-Information like results (Sec. 9.7). However, we chose this form such that, for any given approximation to the original log-likelihood (Eq. 2), the log-likelihood induced by conditioning on the gradient (Eq. 9) has the same form,

$$\log P\left(g_i = g_i | \Lambda_{\text{like},i} = \lambda_{\text{like},i}, w_i^* = w_i^*\right) + \text{const} \approx$$
$$-\frac{\left(g_i - \lambda_{\text{like},i}\left(w_i^* - \mu_i\right)\right)^2}{2\lambda_{\text{like},i}} = -\tfrac{1}{2}\lambda_{\text{like},i}\left(w_i^* - \mu_{\text{like},i}\right). \tag{10}$$

The only material difference between the right-hand-side of this expression and the original likelihood (Eq. 2) is that here the parameter of interest is the optimal weight, $w_i^*$, as opposed to the weight in the underlying generative model, $w_i$. Also note that here we have specified the values for all the random variables: $g_i = g_i$ and $\Lambda_{\text{like},i} = \lambda_{\text{like},i}$ which fix the value of $\mu_{\text{like},i}$ to $\mu_{\text{like}} = \frac{g_i}{\lambda_{\text{like},i}} + \mu_i$.

The key difference between the methods is that in the original problem, where we infer a single distribution over all weights jointly, the underlying weights, $w$, were fixed, and as such, Bayesian filtering reduces to recursively applying Bayes theorem, which does not give rise to interesting dynamics. In contrast, when we consider $N$ separate inference problems, in which $w_i^*$ is inferred from the gradient, then we are forced to introduce dynamics, because $w_i^*$ varies over time, as it depends on all the other parameters in the network, $\mu_{-i}$, which are also being optimized (Fig. 1A). However, the full generative process in Fig. 1A is very difficult to reason about, as it requires us to integrate over all possible trajectories for the other parameters, $\mu_{-i}$. Instead, we write down simplified approximate dynamics over $w_i^*$ directly (Fig. 1B), and evaluate the quality of these simplified dynamics empirically. We choose the model in Fig. 1 such that it has finite-length temporal dependencies, and hence can be written in terms of a Markov model with an expanded state-space. In particular, we define a new random variable, $z_i$, incorporating $w_i^*$, whose generative process is Markovian (Fig. 1C). For Adam, $z_i$ will have two elements representing a parameter, $w_i^*$ and the associated momentum, $p_i$, whereas for RMSprop it will have only one element representing just the parameter,

$$z_{\text{RMSprop},i}(t) = \left(w_i^*(t)\right) \qquad\qquad z_{\text{Adam},i}(t) = \begin{pmatrix} w_i^*(t) \\ p_i(t) \end{pmatrix} \tag{11}$$

As $z_i$ represents the optimal setting for a parameter it will change over time, and we assume a simple Gaussian form for these changes,

$$P\left(z_i(t) | z_i(t-1)\right) = \mathcal{N}\left(z_i(t); (I - A)\,z_i(t-1), Q\right), \tag{12}$$

where $Q$ is the covariance of Gaussian perturbations and $A$ is the dynamics matrix incorporating weight decay and momentum. We can write the second-order Taylor expansion of the likelihood as a function of the expanded latent, $z_i$ (Eq. 11),

$$\log P\left(g_i | w_i^*\right) = \log P\left(g_i | z_i\right) \approx -\tfrac{1}{2}\left(z_i - \mu_{\text{like},i}\right)^T \Lambda_{\text{like},i}\left(z_i - \mu_{\text{like},i}\right), \tag{13}$$

where we have omitted conditioning on $\Lambda_{\text{like},i}$, as in practice $\Lambda_{\text{like},i}$ is estimated from the gradients, and we have omitted time indices for clarity,

$$g_i = g_i(t) \qquad w_i^* = w_i^*(t) \qquad z_i = z_i(t) \qquad \mu_{\text{like},i} = \mu_{\text{like},i}(t) \qquad \Lambda_{\text{like},i} = \Lambda_{\text{like},i}(t).$$

This likelihood is equivalent to our original likelihood (Eq. 10) because the gradients depend only on $w_i^*(t)$. As such, the negative Hessians must take the form,

$$\Lambda_{\text{RMSprop,like},i} = \left(\Lambda_{\text{like},i}\right) \qquad\qquad \Lambda_{\text{Adam,like},i} = \begin{pmatrix} \Lambda_{\text{like},i} & 0 \\ 0 & 0 \end{pmatrix}. \tag{14}$$

## 3 BAYESIAN (KALMAN) FILTERING AS ADAPTIVE SGD

The Gaussian prior and approximate likelihood allows us to use standard two-step Kalman filter updates. First, we propagate the previous time-step's posterior, $\mathrm{P}\left(\boldsymbol{z}_i(t-1)|\mathcal{D}(t-1)\right)$, with mean $\boldsymbol{\mu}_{\mathrm{post},i}(t-1)$ and covariance $\boldsymbol{\Sigma}_{\mathrm{post},i}(t-1)$, forward in time to obtain a prior at time $t$,

$$\mathrm{P}\left(\boldsymbol{z}_i(t)|\mathcal{G}(t-1)\right) = \int d\boldsymbol{z}_i(t-1)\,\mathrm{P}\left(\boldsymbol{z}_i(t)|\boldsymbol{z}_i(t-1)\right)\mathrm{P}\left(\boldsymbol{z}_i(t-1)|\mathcal{G}_i(t-1)\right)$$
$$= \mathcal{N}\left(\boldsymbol{z}_i(t); \boldsymbol{\mu}_{\mathrm{prior},i}(t), \boldsymbol{\Sigma}_{\mathrm{prior},i}(t)\right),$$

where,

$$\boldsymbol{\mu}_{\mathrm{prior},i} = \boldsymbol{\mu}_{\mathrm{prior},i}(t) = (\boldsymbol{I} - \boldsymbol{A})\,\boldsymbol{\mu}_{\mathrm{post},i}(t-1), \tag{15a}$$

$$\boldsymbol{\Sigma}_{\mathrm{prior},i} = \boldsymbol{\Sigma}_{\mathrm{prior},i}(t) = (\boldsymbol{I} - \boldsymbol{A})\,\boldsymbol{\Sigma}_{\mathrm{post},i}(t-1)\,(\boldsymbol{I} - \boldsymbol{A})^T + \boldsymbol{Q}, \tag{15b}$$

and where $\mathcal{G}_i(t-1) = \{g_i(1), g_i(2), \ldots, g_i(t-1)\}$ is all gradients up to time $t-1$. Note that we have also defined abbreviations for $\boldsymbol{\mu}_{\mathrm{prior},i}(t)$ and $\boldsymbol{\Sigma}_{\mathrm{prior},i}(t)$, omitting the temporal index, $t$. Second we use Bayes theorem to incorporate new data,

$$\mathrm{P}\left(\boldsymbol{z}_i(t)|\mathcal{D}(t)\right) = \frac{\mathrm{P}\left(g_i(t)|\boldsymbol{z}_i(t)\right)\mathrm{P}\left(\boldsymbol{z}_i(t)|\mathcal{G}(t-1)\right)}{\mathrm{P}\left(g_i(t)|\mathcal{G}(t-1)\right)} = \mathcal{N}\left(\boldsymbol{z}_i(t); \boldsymbol{\mu}_{\mathrm{post},i}(t), \boldsymbol{\Sigma}_{\mathrm{post},i}(t)\right).$$

where,

$$\boldsymbol{\Sigma}_{\mathrm{post},i} = \boldsymbol{\Sigma}_{\mathrm{post},i}(t) = \left(\boldsymbol{\Sigma}_{\mathrm{prior},i}^{-1} + \boldsymbol{\Lambda}_{\mathrm{like},i}\right)^{-1}, \tag{16a}$$

$$\boldsymbol{\mu}_{\mathrm{post},i} = \boldsymbol{\mu}_{\mathrm{post},i}(t) = \boldsymbol{\mu}_{\mathrm{prior},i} + \boldsymbol{\Sigma}_{\mathrm{post},i}\boldsymbol{\Lambda}_{\mathrm{like},i}\left(\boldsymbol{\mu}_{\mathrm{like},i} - \boldsymbol{\mu}_{\mathrm{prior},i}\right). \tag{16b}$$

Thus far, we have simply repeated standard Kalman filtering results. To relate Kalman filtering to gradient ascent, we compute the gradient of Eq. (13) at $\boldsymbol{z}_i = \boldsymbol{\mu}_{\mathrm{prior},i}$,

$$\boldsymbol{g}_i = \left.\frac{\partial \log \mathrm{P}\left(g_i|\boldsymbol{z}\right)}{\partial \boldsymbol{z}_i}\right|_{\boldsymbol{z}_i=\boldsymbol{\mu}_{\mathrm{prior},i}} = \boldsymbol{\Lambda}_{\mathrm{like}}\left(\boldsymbol{\mu}_{\mathrm{like},i} - \boldsymbol{\mu}_{\mathrm{prior},i}\right). \tag{17}$$

Note that as the log-likelihood depends only on $w_i^*$, we have,

$$\boldsymbol{g}_{\mathrm{RMSprop},i} = (g_i) \qquad\qquad \boldsymbol{g}_{\mathrm{Adam},i} = \begin{pmatrix} g_i \\ 0 \end{pmatrix}. \tag{18}$$

Now, we identify this gradient (Eq. 17) in the mean updates (Eq. 16b),

$$\boldsymbol{\mu}_{\mathrm{post},i} = \boldsymbol{\mu}_{\mathrm{prior},i} + \boldsymbol{\Sigma}_{\mathrm{post},i}\,\boldsymbol{g}_i. \tag{19}$$

This form is extremely intuitive: it states that the uncertainty should be used to precondition gradient updates, such that the updates are larger when there is more uncertainty, and smaller when past data gives high confidence in the current estimates.

As the precision is always rank-1 (Eq. 14), we can always write it as,

$$\boldsymbol{\Lambda}_{\mathrm{like},i} = \boldsymbol{e}_i\boldsymbol{e}_i^T, \qquad \text{where} \qquad \boldsymbol{e}_{\mathrm{RMSprop},i} = (e_i) \qquad \boldsymbol{e}_{\mathrm{Adam},i} = \begin{pmatrix} e_i \\ 0 \end{pmatrix} \tag{20}$$

As such, the updates for the posterior covariance (Eq. 16a) can be rewritten using the Sherman Morrison formula (Hager, 1989),

$$\boldsymbol{\Sigma}_{\mathrm{post},i} = \boldsymbol{\Sigma}_{\mathrm{prior},i} - \frac{\boldsymbol{\Sigma}_{\mathrm{prior},i}\boldsymbol{e}_i\boldsymbol{e}_i^T\boldsymbol{\Sigma}_{\mathrm{prior},i}}{1 + \boldsymbol{e}_i^T\boldsymbol{\Sigma}_{\mathrm{prior},i}\boldsymbol{e}_i}. \tag{21}$$

To estimate $\boldsymbol{e}_i$, we use the Fisher information (see SI Sec. 9.7). In particular, we could use the gradient itself, but could also (under weaker approximations) use the centred gradient (Graves, 2013), or the gradient for data sampled under the prior.

## 4    BAYESIAN RMSPROP (BRMSPROP)

Here, we develop a Bayesian variant of RMSprop, which we call BRMSprop. We consider each parameter to be inferred by considering a separate Bayesian inference problem, so the latent variable, $z = w^*$, is a single scalar, representing a single parameter (we omit the index $i$ on $z$ for brevity). We use $A = \eta^2/(2\sigma^2)$ and $Q = \eta^2$ giving a dynamical prior,

$$\mathrm{P}\left(z(t+1)|z(t)\right) = \mathcal{N}\left(\left(1 - \tfrac{\eta^2}{2\sigma^2}\right)z(t), \eta^2\right), \tag{22}$$

such that the stationary distribution over $z$ has standard-deviation $\sigma$. To obtain a complete description of our updates, we substitute these choices into the updates for the prior (Eq. 15) and the posterior (Eq. 19 and Eq. 21),

$$\mu_{\mathrm{prior}} = \left(1 - \tfrac{\eta^2}{2\sigma^2}\right)\mu_{\mathrm{post}}(t-1) \qquad\qquad \mu_{\mathrm{post}} = \mu_{\mathrm{prior}} + \sigma_{\mathrm{post}}^2 g, \tag{23}$$

$$\sigma_{\mathrm{prior}}^2 = \left(1 - \tfrac{\eta^2}{2\sigma^2}\right)^2 \sigma_{\mathrm{post}}^2(t-1) + \eta^2 \qquad\qquad \sigma_{\mathrm{post}}^2 = \sigma_{\mathrm{prior}}^2 \left(1 - \frac{\sigma_{\mathrm{prior}}^2 e^2}{1 + \sigma_{\mathrm{prior}}^2 e^2}\right) \tag{24}$$

For an efficient implementation of the full algorithm, see SI Alg. 1.

### 4.1    RECOVERING RMSPROP FROM BRMSPROP

Now we show that with in steady state, BRMSprop closely approximates RMSprop. Making this comparison is non-trivial because the "additional" variables in RMSprop and BRMSprop (i.e. the average squared gradient and the uncertainty respectively) are not directly comparable. However, the implied learning rate *is* directly comparable. We therefore look at the learning rate when the average squared gradient and the uncertainty have reached steady-state. As $t \to \infty$, we expect $\sigma_{\mathrm{post}}^2$ to reach steady-state, at which point, $\sigma_{\mathrm{post}}^2 = \sigma_{\mathrm{post}}^2(t) = \sigma_{\mathrm{post}}^2(t+1)$. In SI Sec. 9.1, we consider steady-state in the general case. Applying those results to RMSprop, we obtain,

$$\sigma_{\mathrm{post}}^4(t+1)\langle e^2\rangle \approx \eta^2.$$

Solving for $\sigma_{\mathrm{post}}^2$,

$$\sigma_{\mathrm{post}}^2 \approx \frac{\eta}{\sqrt{\langle e^2\rangle}}.$$

Substituting this form into Eq. (23) recovers the root-mean-square normalization used in RMSprop.

## 5    BAYESIAN ADAM (BADAM)

We now develop a Bayesian variant of Adam (Graves, 2013; Kingma & Ba, 2015), which we call BAdam. To introduce momentum into our Bayesian updates, we introduce an auxiliary momentum variable, $\mathrm{p}(t)$, corresponding to each parameter, $\mathrm{w}^*(t)$,

$$\mathbf{z}(t) = \begin{pmatrix} \mathrm{w}^*(t) \\ \mathrm{p}(t) \end{pmatrix},$$

then we infer both the parameter and momentum jointly. Under the prior, the momentum, $\mathrm{p}(t)$, evolves through time independently of $\mathrm{w}^*(t)$, obeying an AR(1) (or equivalently a discretised Ornstein-Uhlenbeck) process, with decay $\eta_{\mathrm{p}}$ and injected noise variance $\eta_{\mathrm{p}}^2$,

$$\mathrm{p}(t+1) = (1 - \eta_{\mathrm{p}})\,\mathrm{p}(t) + \eta_{\mathrm{p}}\,\xi_{\mathrm{p}}(t), \tag{25}$$

where $\xi_{\mathrm{p}}(t)$ is standard Gaussian noise. This particular coupling of the injected noise variance and the decay rate ensures that the influence of the momentum on the weight is analogous to unit-variance Gaussian noise (see SI Sec. 9.2). The weight obeys similar dynamics, with the addition of a momentum-dependent term which in practice causes changes in $\mathrm{w}_i^*(t)$ to be correlated across time (i.e. multiple consecutive increases or decreases in $\mathrm{w}_i^*(t)$),

$$\mathrm{w}^*(t+1) = \left(1 - \tfrac{\eta^2 + \eta_{\mathrm{w}}^2}{2\sigma^2}\right)\mathrm{w}^*(t) + \eta\mathrm{p}(t) + \eta_{\mathrm{w}}\,\xi_{\mathrm{w}}(t), \tag{26}$$

where $\xi_w(t)$ is again standard Gaussian noise, $\eta$ is the strength of the momentum coupling, $\frac{\eta^2 + \eta_w^2}{2\sigma^2}$ is the strength of the weight decay, and $\eta_w^2$ is the variance of the noise injected to the weight. It is possible to write these dynamics in the generic form given above (Eq. 12), by using,

$$\boldsymbol{A} = \begin{pmatrix} \frac{\eta_w^2 + \eta^2}{2\sigma^2} & -\eta \\ 0 & \eta_p \end{pmatrix} \qquad \boldsymbol{Q} = \begin{pmatrix} \eta_w^2 & 0 \\ 0 & \eta_p^2 \end{pmatrix}, \tag{27}$$

and these settings fully determine the updates, according to Eq. (15) and Eq. (16). For an efficient implementation of the full algorithm, see SI Alg. 2.

## 5.1 RECOVERING ADAM FROM BADAM

Now we show that with suitable choices for the parameters, BAdam closely approximates Adam. In particular, we compare the updates for the (posterior) parameter estimate, $\mu_w$, and the momentum, $\mu_p$, when we eliminate weight decay by setting $\sigma^2 = \infty$, and eliminate noise injected directly into the weight by setting $\eta_w = 0$,

$$\mu_w(t+1) = \mu_{\text{post,w}}(t+1) = \mu_w(t) + \eta\mu_p(t) + \Sigma_{ww}g(t), \tag{28a}$$
$$\mu_p(t+1) = \mu_{\text{post,p}}(t+1) = (1 - \eta_p)\,\mu_p(t) + \Sigma_{wp}g(t). \tag{28b}$$

These updates depend on two quantities, $\Sigma_{ww}$ and $\Sigma_{wp}$, which are related to $\langle e^2 \rangle$, but have no direct analogue in standard Adam. As such, to make a direct comparison, we use the same approach as we used previously for RMSprop: we compare the updates for the parameter and momentum, when $\Sigma_{ww}$, $\Sigma_{wp}$ in BAdam and $\langle e^2 \rangle$ in Adam have reached their steady-state values. To find the steady-states for $\Sigma_{ww}$ and $\Sigma_{wp}$, we again use the simplified covariance updates derived in SI Sec. 9.1,

$$\boldsymbol{Q} \approx \boldsymbol{A\Sigma} + \boldsymbol{\Sigma A}^T + \boldsymbol{\Sigma}\langle \boldsymbol{ee}^T \rangle \boldsymbol{\Sigma}. \tag{29}$$

Substituting Eq. (27) and Eq. (20) into Eq. (29), and again using $\sigma^2 = \infty$ and $\eta_w^2 = 0$, we obtain,

$$\begin{pmatrix} 0 & 0 \\ 0 & \eta_p^2 \end{pmatrix} \approx \eta_p \begin{pmatrix} 0 & \Sigma_{wp} \\ \Sigma_{wp} & 2\Sigma_{pp} \end{pmatrix} - \eta \begin{pmatrix} 2\Sigma_{wp} & \Sigma_{pp} \\ \Sigma_{pp} & 0 \end{pmatrix} + \langle e^2 \rangle \begin{pmatrix} \Sigma_{ww}^2 & \Sigma_{ww}\Sigma_{wp} \\ \Sigma_{ww}\Sigma_{wp} & \Sigma_{wp}^2 \end{pmatrix}. \tag{30}$$

We now assume that the data is informative, in the sense that it is strong enough to give a narrow posterior relative to the prior (without which any neural network training algorithm is unlikely to be able to obtain good performance). This implies $\Sigma_{pp} \ll \eta_p/2$ (see SI Sec. 9.2), allowing us to solve for $\Sigma_{wp}$, using the lower-right element of the above expression (i.e. $\eta_p^2 \approx 2\eta_p\Sigma_{pp} + \langle e^2 \rangle \Sigma_{wp}^2$),

$$\Sigma_{wp} \approx \frac{\eta_p}{\sqrt{\langle e^2 \rangle}}. \tag{31}$$

To recover a very close approximation to Adam, we need a specific relationship between learning rates and evidence strength, such that,

$$\Sigma_{ww} \approx \frac{\eta\eta_p}{\sqrt{\langle e^2 \rangle}}.$$

While this may seem restrictive, it is only necessary to achieve the closest possible match between Bayesian filtering and plain Adam. However, our goal is not to match Adam exactly, given that Adam does not even converge (Reddi et al., 2018). Instead, our goal is to capture the essential dynamical insights of Adam in a Bayesian method. Indeed, we hope that the resulting Bayesian method constitutes an improvement over Adam, which implies that it must exhibit some differences. Nonetheless, focusing on the regime where filtering and Adam are most similar, the filtering updates become,

$$\mu_w(t+1) \approx \mu_w(t) + \eta\mu_p(t) + \eta\eta_p \frac{g(t)}{\sqrt{\langle e^2 \rangle}} \approx \mu_w(t) + \eta\mu_p(t+1) \tag{32a}$$

$$\mu_p(t+1) \approx (1 - \eta_p)\,\mu_p(t) + \eta_p \frac{g(t)}{\sqrt{\langle e^2 \rangle}}, \tag{32b}$$

where we have substituted for $\mu_p(t+1)$ into Eq. 32a, which assumes that $\eta_p \ll 1$. This is very close to the Adam updates, except that the root-mean-square normalization is in the momentum updates,

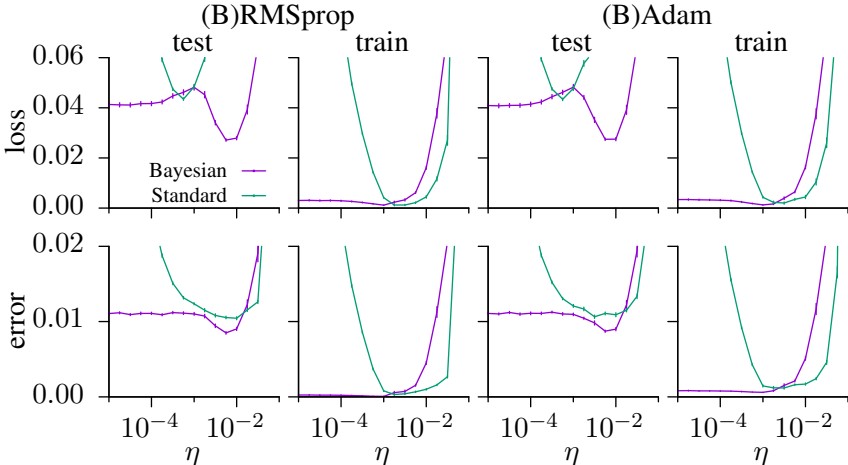

Figure 2: Comparing RMSprop and BRMSprop on MNIST. The vertical bars represent one standard error and are based on 30 independent runs. Adam and BAdam both use $\eta_w = 0.1$.

rather than the parameter updates. To move the location of the root-mean-square normalization, we rewrite the updates in terms of a rescaled momentum,

$$\tilde{\mu}_p(t+1) = \mu_p(t+1)\sqrt{\langle e^2 \rangle} \tag{33}$$

giving,

$$\mu_w(t+1) \approx \mu_w(t) + \eta \frac{\tilde{\mu}_p(t+1)}{\sqrt{\langle e^2 \rangle}}, \tag{34a}$$

$$\tilde{\mu}_p(t+1) \approx (1 - \eta_p)\,\tilde{\mu}_p(t) + \eta_p g(t), \tag{34b}$$

which recovers Adam. See SI Sec. 9.6 for a similar discussion for NAG/NAdam

## 6 EXPERIMENTS

We compared Bayesian and standard methods on MNIST. In particular, we trained a CNN with relu nonlinearities and maxpooling for 50 epochs. The network had two convolutional layers with 10 channels in the first layer and 20 in the second, with $5 \times 5$ convolutional kernels, followed by a single fully connected layer with 50 units, and was initialized with draws from a Gaussian with variance $2/N_{\text{inputs}}$. The network did not use dropout (or any other form of stochastic regularisation), which would increase the variance of the gradients, artificially inflating $\langle e^2 \rangle$.

The key Bayesian-specific parameters are the variance of the stationary distribution and the initial uncertainty; both of which were set to $1/(2N_{\text{inputs}})$. In principle, they should be similar to the initialization variance, but in practice we found that using a somewhat lower value gave better performance, though this needs further investigation. For the RMSprop and Adam specific parameters, we used the PyTorch defaults.

Comparing the Bayesian and non-Bayesian methods, we do not expect to see very large discrepancies, because they are approximately equivalent in steady-state. Nonetheless, the Bayesian methods show considerably lower test loss, somewhat lower classification error, and similar training loss and training error (Fig. 2). The local maximum in test-loss may arise because we condition on each datapoint multiple times, which is not theoretically justified (correcting this is non-trivial in the dynamical setting, so we leave it for future work).

## 7 FEATURES OF BAYESIAN STOCHASTIC GRADIENT DESCENT

Here we summarise the features of Bayesian stochastic gradient descent schemes, pointing out where we recover previously known best practice and where our approach suggests new ideas.

### 7.1 WEIGHT DECAY, L2 REGULARIZATION AND BAYESIAN PRIORS

Loshchilov & Hutter (2017) recently examined different forms of weight decay in adaptive SGD methods such as Adam. In particular, they asked whether or not the root-mean-square normalizer should be applied to the weight decay term. In common practice, we take weight decay as arising from the gradient of an L2 regularizer, in which case it is natural to normalize the gradient of the objective and the gradient of the regularizer in the same way. However, there is an alternative: to normalize only the gradient of the objective, but to keep the weight decay constant (in which case, weight decay cannot be interpreted as arising from an L2 normalizer). Loshchilov & Hutter (2017) show that this second method, which they call AdamW, gives better test accuracy than the standard approach. Remarkably, AdamW arises naturally in BRMSprop and BAdam (e.g. Eq. 23), providing a potential explanation for its improved performance.

### 7.2 NESTEROV ACCELERATED GRADIENTS/WEIGHT DECAY

In Nesterov accelerated gradients (NAG), we compute the gradient at a "predicted" location formed by applying a momentum update to the current parameters (Nesterov, 1983). These updates arise naturally in our Bayesian scheme, as the required gradient term (Eq. 17) is evaluated at $\boldsymbol{\mu}_{\text{prior}}$ (Eq. 15a), which is precisely a prediction formed by combining the current setting of the parameters, $\boldsymbol{\mu}_{\text{post}}(t)$, with momentum and decay, embodied in $\boldsymbol{A}$ (Eq. 27) to form a prediction, $\boldsymbol{\mu}_{\text{prior}}(t+1)$. It should be noted that the original Nesterov acceleration (Nesterov, 1983) had no gradient preconditioning. Instead, our approach corresponds to NAdam (Dozat, 2016). Interestingly, as we also implement weight decay through the dynamics matrix, $\boldsymbol{A}$, we should also apply the updates from weight decay before computing the gradient, giving, to our knowledge, a novel method that we denote "Nesterov accelerated weight decay" (NAWD).

### 7.3 CONVERGENCE

A series of recent papers have discussed the convergence properties of RMSprop and Adam, noting that they may fail to converge (Wilson et al., 2017; Reddi et al., 2018) if the exponentially decaying average over the squared gradients is computed with a too-small timescale (Reddi et al., 2018).

Our method circumvents these issues by coupling the learning rate to the timescale over which the implicit exponential moving average for the mean-square gradient is performed. As such, in the limit as the learning rates go to zero (i.e. $\eta \to 0$ for BRMSprop and $\eta_{\text{w}} \to 0$ and $\eta_{\text{p}} \to 0$ for BAdam), our method becomes SGD with adaptive learning rates that scale as $1/t$ and is therefore likely to be convergent for convex functions (Robbins & Monro, 1951; Bottou, 1998), though we leave a rigorous proof to future work.

### 7.4 UPDATING THE PRECONDITIONER BEFORE APPLYING THE UPDATE

ADAM (Kingma & Ba, 2015) first updates the root-mean-square gradient normalizer before computing the parameter update. This is important, because it ensures that updates are bounded in the pathological case that gradients are initially all very small, such that the root-mean-square normalizer is also small, and then there is a single large gradient. Bayesian filtering naturally recovers this choice as the gradient preconditioner in Eq. (19) is the posterior, rather than the prior, covariance (i.e. updated with the current gradient).

## 8 DISCUSSION

Here, we developed BRMSprop and BAdam, Bayesian versions of RMSprop and Adam respectively. These methods naturally recover many features of state-of-the-art adaptive SGD methods, including the root-mean-square normalizer, Neseterov acceleration and AdamW. As such, BRMSprop and BAdam provide a possible explanation for the empirical success of RMSprop, Adam, Nesterov acceleration and AdamW. Experimentally, we find, BRMSprop and BAdam are superior to the standard methods in the tasks that we examined. Finally, the novel interpretation of adaptive SGD as filtering in a dynamical model opens several avenues for future research, including new stochastic regularisation methods and combinations with Kronecker factorisation.

We believe that it should be possible to combine the temporal changes induced by the factorised approximations with other types of inference, including probabilistic backpropagation (Hernández-Lobato & Adams, 2015) or assumed density filtering (Ghosh et al., 2016). At present, our method bears closest relation to natural gradient variational inference methods (Zhang et al., 2017; Khan et al., 2018), as they also use the Fisher Information to approximate the likelihood. Indeed, our method becomes equivalent to these approaches in the limit as we send the learning rate, $\eta$ to zero. The key difference is that because they do not formulate a factorised generative model, they are unable to provide a strong justification for the introduction of rich dynamics, and they are unable to reason about optimal inference under these dynamics.

Bayesian filtering presents a novel approach to neural network optimization, and as such, there are variety of directions for future work. First, Bayesian filtering converts the problem of neural network optimization into the statistical problem of understanding the dynamics of changes in the optimal weight induced by optimization in the other parameters. In particular, we can perform an empirical investigation in large scale systems, or attempt to find closed-form expressions for the dynamics in simplified domains such as linear regression. Second, here we wrote down a statistical model for the gradient. However, there are many circumstances where the gradient is not available. Perhaps a low precision or noisy gradient is available due to noise in the parameters (e.g. due to dropout Srivastava et al., 2014), or perhaps we wish to consider a biological setting, where the gradient is not present at all (Aitchison et al., 2014). The Bayesian approach presented here gives a straightforward recipe for developing (Bayes) optimal algorithms for such problems. Third, stochastic regularization has been shown to be extremely effective at reducing generalization error in neural networks. This Bayesian interpretation of adaptive SGD methods presents opportunities for new stochastic regularization schemes. Fourth, it should be possible to develop filtering methods that represent the covariance of a full weight matrix by exploiting Kronecker factorisation (Martens & Grosse, 2015; Grosse & Martens, 2016; Zhang et al., 2017).

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

## 9 SUPPLEMENTARY INFORMATION

### 9.1 STEADY-STATE COVARIANCE

To understand the steady-state behaviour of the covariance, we begin by substituting Eq. 21 into Eq. 15b to form a combined update,

$$\boldsymbol{\Sigma}_{\text{prior}}(t+1) = (\boldsymbol{I} - \boldsymbol{A})\left(\boldsymbol{\Sigma}_{\text{prior}}(t) - \frac{\boldsymbol{\Sigma}_{\text{prior}}(t)\boldsymbol{e}\boldsymbol{e}^T\boldsymbol{\Sigma}_{\text{prior}}(t)}{1 + \boldsymbol{e}^T\boldsymbol{\Sigma}_{\text{prior}}(t)\boldsymbol{e}^T}\right)(\boldsymbol{I} - \boldsymbol{A})^T + \boldsymbol{Q}. \tag{35}$$

Note that this gives updates for $\boldsymbol{\Sigma}_{\text{prior}}(t)$, rather than $\boldsymbol{\Sigma}_{\text{post}}(t)$, as this simplifies the algebra, but we will move back to $\boldsymbol{\Sigma}_{\text{post}}(t)$ at the end. As $t \to \infty$, we reach steady state for which, $\boldsymbol{\Sigma}_{\text{prior}} = \boldsymbol{\Sigma}_{\text{prior}}(t) = \boldsymbol{\Sigma}_{\text{prior}}(t+1)$. Substituting this asympototic covariance into Eq. (35) and rearranging, we obtain,

$$(\boldsymbol{I} - \boldsymbol{A})\left(\boldsymbol{\Sigma}_{\text{prior}}\boldsymbol{e}\boldsymbol{e}^T\boldsymbol{\Sigma}_{\text{prior}}\right)(\boldsymbol{I} - \boldsymbol{A})^T = \left(\boldsymbol{A}\boldsymbol{\Sigma}_{\text{prior}}\boldsymbol{A}^T - \boldsymbol{A}\boldsymbol{\Sigma}_{\text{prior}} - \boldsymbol{\Sigma}_{\text{prior}}\boldsymbol{A}^T + \boldsymbol{Q}\right)\left(1 + \boldsymbol{e}^T\boldsymbol{\Sigma}_{\text{prior}}\boldsymbol{e}^T\right) \tag{36}$$

Now our goal is to find the asymptotic scaling of $\boldsymbol{\Sigma}_{\text{prior}}$ with $\eta$ so as to obtain a valid approximation to this steady-state expression. To begin, we know that,

$$\boldsymbol{A} \in \mathcal{O}\left(\eta\right) \qquad\qquad \boldsymbol{Q} \in \Theta\left(\eta^2\right),$$

where $\mathcal{O}$ denotes an asymptotic upper bound, and $\Theta$ denotes an asymptotic bound on both sides (Knuth, 1976). Here, $\eta$ is the learning rate, which is assumed to be small, and hence we take the limit of $\eta \to 0$ for the purposes of the limits in $\mathcal{O}$ and $\Theta$. Note that for Adam we have $\boldsymbol{A}_{\text{Adam}} \in \Theta(\eta)$ but for RMSprop we have $\boldsymbol{A}_{\text{RMSprop}} \in \Theta(\eta^2)$, and we summarise both using the asymptotic upper bound above.

The right-hand-side of Eq. (36) is thus bounded above,

$$\left(\boldsymbol{A}\boldsymbol{\Sigma}_{\text{prior}}\boldsymbol{A}^T - \boldsymbol{A}\boldsymbol{\Sigma}_{\text{prior}} - \boldsymbol{\Sigma}_{\text{prior}}\boldsymbol{A}^T + \boldsymbol{Q}\right)\left(1 + \boldsymbol{e}^T\boldsymbol{\Sigma}_{\text{prior}}\boldsymbol{e}^T\right) \in \mathcal{O}\left(\eta^2 + \eta\Sigma(\eta) + \eta\Sigma^2(\eta)\right) \tag{37}$$

where $\Sigma(\eta)$ describes the scaling of $\boldsymbol{\Sigma}_{\text{prior}}$,

$$\boldsymbol{\Sigma}_{\text{prior}} \in \Theta(\Sigma(\eta)).$$

The left-hand-side can be asymptotically bounded on both sides as $\lim_{\eta \to 0} \boldsymbol{A} = \boldsymbol{0}$,

$$(\boldsymbol{I} - \boldsymbol{A}) \left(\boldsymbol{\Sigma}_{\text{prior}} \boldsymbol{e} \boldsymbol{e}^T \boldsymbol{\Sigma}_{\text{prior}}\right) (\boldsymbol{I} - \boldsymbol{A})^T \in \Theta \left(\Sigma^2(\eta)\right). \tag{38}$$

The asymptotic upper bound (Eq. 37) for the right-hand-side of Eq. (36) must bound the asymptotic lower bound for the left-hand-side (Eq. 38),

$$\Sigma^2(\eta) \in \mathcal{O}\left(\eta^2 + \eta\Sigma(\eta) + \eta\Sigma^2(\eta)\right). \tag{39}$$

Now, we consider the case in which each of the terms $\eta^2$, $\eta\Sigma(\eta)$ or $\eta\Sigma^2(\eta)$ is asymptotically dominant (potentially along with other terms). The $\eta\Sigma^2(\eta)$ term cannot dominate, because it cannot form an asymptotic upper bound on $\Sigma^2(\eta)$. As such, $\Sigma^2(\eta)$ could be asymptotically bound either by $\eta^2$, or by $\eta\Sigma(\eta)$ (or both simultaneously). If it is bound by $\eta^2$, we have,

$$\Sigma^2(\eta) \in \mathcal{O}\left(\eta^2\right) \qquad \Longrightarrow \qquad \Sigma(\eta) \in \mathcal{O}\left(\eta\right) \tag{40}$$

Alternatively, if it is bound by $\eta\Sigma(\eta)$, we have,

$$\Sigma^2(\eta) \in \mathcal{O}\left(\eta\Sigma(\eta)\right) \qquad \Longrightarrow \qquad \Sigma(\eta) \in \mathcal{O}\left(\eta\right) \tag{41}$$

And as it must be bound by one (or both) of these terms, we know that $\Sigma(\eta)$ and hence $\boldsymbol{\Sigma}_{\text{prior}}$ are bounded by $\eta$,

$$\Sigma(\eta) \in \mathcal{O}\left(\eta\right) \qquad \Longrightarrow \qquad \boldsymbol{\Sigma}_{\text{prior}} \in \mathcal{O}\left(\eta\right). \tag{42}$$

Now that we have the scaling of $\boldsymbol{\Sigma}_{\text{prior}}$, deriving a simplified approximation to Eq. (36) is straightforward. In particular, we consider only terms that scale with $\eta^2$, neglecting those that scale with $\eta^3$. We now consider terms to be eliminated in Eq. (36). In particular, on the left-hand-side,

$$\boldsymbol{A}\left(\boldsymbol{\Sigma}_{\text{prior}} \boldsymbol{e} \boldsymbol{e}^T \boldsymbol{\Sigma}_{\text{prior}}\right) \in \mathcal{O}(\eta^3), \tag{43a}$$

$$\left(\boldsymbol{\Sigma}_{\text{prior}} \boldsymbol{e} \boldsymbol{e}^T \boldsymbol{\Sigma}_{\text{prior}}\right) \boldsymbol{A}^T \in \mathcal{O}(\eta^3), \tag{43b}$$

$$\boldsymbol{A}\left(\boldsymbol{\Sigma}_{\text{prior}} \boldsymbol{e} \boldsymbol{e}^T \boldsymbol{\Sigma}_{\text{prior}}\right) \boldsymbol{A}^T \in \mathcal{O}(\eta^4), \tag{43c}$$

and on the right-hand-side,

$$\boldsymbol{A}\boldsymbol{\Sigma}_{\text{prior}} \boldsymbol{A}^T \left(1 + \boldsymbol{e}^T \boldsymbol{\Sigma}_{\text{prior}} \boldsymbol{e}\right) \in \mathcal{O}(\eta^3), \tag{43d}$$

$$\left(\boldsymbol{Q} - \boldsymbol{A}\boldsymbol{\Sigma}_{\text{prior}} - \boldsymbol{\Sigma}_{\text{prior}} \boldsymbol{A}^T\right) \left(\boldsymbol{e}^T \boldsymbol{\Sigma}_{\text{prior}} \boldsymbol{e}\right) \in \mathcal{O}(\eta^3). \tag{43e}$$

Neglecting these terms, and instead focusing only on the $\mathcal{O}(\eta^2)$ terms we obtain,

$$\boldsymbol{\Sigma}_{\text{prior}} \boldsymbol{e} \boldsymbol{e}^T \boldsymbol{\Sigma}_{\text{prior}} \approx \boldsymbol{Q} - \boldsymbol{A}\boldsymbol{\Sigma}_{\text{prior}} - \boldsymbol{\Sigma}_{\text{prior}} \boldsymbol{A}^T. \tag{44}$$

However, for the purposes of our updates, we are interested in the asymptotic behaviour of $\boldsymbol{\Sigma}_{\text{post}}$, not $\boldsymbol{\Sigma}_{\text{prior}}$. We therefore substitute for $\boldsymbol{\Sigma}_{\text{prior}}$ from Eq. (15b) in Eq. (44), and neglect third-order terms. First, we consider the the left-hand-side of Eq. (44),

$$\boldsymbol{\Sigma}_{\text{prior}} \boldsymbol{e} \boldsymbol{e}^T \boldsymbol{\Sigma}_{\text{prior}} = \left((\boldsymbol{I} - \boldsymbol{A}) \boldsymbol{\Sigma}_{\text{post}} (\boldsymbol{I} - \boldsymbol{A}) + \boldsymbol{Q}\right) \boldsymbol{e} \boldsymbol{e}^T \left((\boldsymbol{I} - \boldsymbol{A}) \boldsymbol{\Sigma}_{\text{post}} (\boldsymbol{I} - \boldsymbol{A}) + \boldsymbol{Q}\right)$$
$$\approx \boldsymbol{\Sigma}_{\text{post}} \boldsymbol{e} \boldsymbol{e}^T \boldsymbol{\Sigma}_{\text{post}}. \tag{45}$$

And we do the same for the right-hand-side of Eq (44),

$$\boldsymbol{Q} - \boldsymbol{A}\boldsymbol{\Sigma}_{\text{prior}} - \boldsymbol{\Sigma}_{\text{prior}} \boldsymbol{A}^T = \boldsymbol{Q} - \boldsymbol{A}\left((\boldsymbol{I} - \boldsymbol{A}) \boldsymbol{\Sigma}_{\text{post}} (\boldsymbol{I} - \boldsymbol{A}) + \boldsymbol{Q}\right) - \left((\boldsymbol{I} - \boldsymbol{A}) \boldsymbol{\Sigma}_{\text{post}} (\boldsymbol{I} - \boldsymbol{A}) + \boldsymbol{Q}\right) \boldsymbol{A}^T$$
$$\approx \boldsymbol{Q} - \boldsymbol{A}\boldsymbol{\Sigma}_{\text{post}} - \boldsymbol{\Sigma}_{\text{post}} \boldsymbol{A}^T \tag{46}$$

Combining these approximations to the left and right hand sides, we obtain the result used in the main text for Adam.

$$\boldsymbol{\Sigma}_{\text{post}} \boldsymbol{e} \boldsymbol{e}^T \boldsymbol{\Sigma}_{\text{post}} \approx \boldsymbol{Q} - \boldsymbol{A}\boldsymbol{\Sigma}_{\text{post}} - \boldsymbol{\Sigma}_{\text{prior}} \boldsymbol{A}^T. \tag{47}$$

For RMSprop, remember that $\boldsymbol{A}_{\text{RMSprop}} \in \mathcal{O}(\eta^2)$, so $\boldsymbol{A}\boldsymbol{\Sigma}_{\text{post}} \in \mathcal{O}(\eta^3)$, and we can obtain an even simpler result,

$$\boldsymbol{\Sigma}_{\text{post}} \boldsymbol{e} \boldsymbol{e}^T \boldsymbol{\Sigma}_{\text{post}} \approx \boldsymbol{Q}. \tag{48}$$

These equations bear a remarkable resemblance to the continuous time Kalman filter that emerges when you take the limit of small timesteps (Xie et al., 2007).

## 9.2 Setting the momentum decay

In the main text, we briefly note that we couple the injected noise and decay in Eq. (25) such that the effect of $p(t)$ on $w(t)$ is analogous to that of unit-variance Gaussian noise. In particular, we require that the stationary variance of $w(t)$ is again $\sigma^2$, despite the momentum term in Eq. (26) being treated as if it was unit-variance Gaussian noise. To show that the stationary variance of $w(t)$ is indeed $\sigma^2$, we solve the Lyapunov equation for the stationary covariance, $\boldsymbol{\Psi}$,

$$\boldsymbol{\Psi} = (\boldsymbol{I} - \boldsymbol{A}) \, \boldsymbol{\Psi} \, (\boldsymbol{I} - \boldsymbol{A})^T + \boldsymbol{Q},$$

where we do not condition on data. Neglecting the second-order term in $\boldsymbol{A}$,

$$\boldsymbol{Q} \approx \boldsymbol{A}\boldsymbol{\Psi} + (\boldsymbol{A}\boldsymbol{\Psi})^T, \tag{49}$$

and as,

$$\boldsymbol{A}\boldsymbol{\Psi} = \begin{pmatrix} \frac{\eta^2 + \eta_w^2}{2\sigma^2} & -\eta \\ 0 & \eta_p \end{pmatrix} \begin{pmatrix} \Psi_{ww} & \Psi_{wp} \\ \Psi_{wp} & \Psi_{pp} \end{pmatrix} = \begin{pmatrix} \frac{\eta^2 + \eta_w^2}{2\sigma^2} \Psi_{ww} - \eta \Psi_{wp} & \frac{\eta^2 + \eta_w^2}{2\sigma^2} \Psi_{wp} - \eta \Psi_{pp} \\ \eta_p \Psi_{wp} & \eta_p \Psi_{pp} \end{pmatrix}.$$

Eq. (49) becomes,

$$\begin{pmatrix} \eta_w^2 & 0 \\ 0 & \eta_p^2 \end{pmatrix} \approx \begin{pmatrix} 2 \left( \frac{\eta^2 + \eta_w^2}{2\sigma^2} \Psi_{ww} - \eta \Psi_{wp} \right) & \frac{\eta^2 + \eta_w^2}{2\sigma^2} \Psi_{wp} - \eta \Psi_{pp} + \eta_p \Psi_{wp} \\ \frac{\eta^2 + \eta_w^2}{2\sigma^2} \Psi_{wp} - \eta \Psi_{pp} + \eta_p \Psi_{wp} & 2\eta_p \Psi_{pp} \end{pmatrix} \tag{50}$$

Now, we can solve for $\Psi_{pp}$ using the ppth element of Eq. (50),

$$\Psi_{pp} = \frac{\eta_p}{2},$$

and we can solve for $\Psi_{wp}$ using the wpth element of Eq. (50), in combination with our solution to $\Psi_{pp}$,

$$\Psi_{wp} = \frac{\eta \eta_p / 2}{\eta_p + \frac{\eta^2 + \eta_w^2}{2\sigma^2}} \approx \eta/2.$$

This approximation becomes exact in the limit we consider to recover Adam (i.e. $\sigma^2 = \infty$), but it also holds when $\sigma^2$ is finite, because $\eta$, $\eta_w$ and $\eta_p$ are all smaller than 1, and $\eta_p$ — the "learning rate" for the momentum — is usually larger than $\eta$ and $\eta_w$ (in fact, $\eta_p$ can be as high as $0.1$). Finally, we can solve for $\Psi_{ww}$ using the wwth element of Eq. (50),

$$\Psi_{ww} \approx \sigma^2,$$

as required.

## 9.3 Efficient implementation of BRMSprop

See Alg. 1 for a complete specification of our approach. There are two important points to note. First, the method is as memory efficient as RMSprop, as the updates for $\Sigma$ and $\mu$ can be done in-place, so there is no need to maintain separate $\mu_{\text{prior}}$ and $\mu_{\text{post}}$ and $\sigma^2_{\text{prior}}$ and $\sigma^2_{\text{post}}$ in memory. Second, we simplified the covariance updates in Eq. (24),

$$\sigma^2_{\text{post}} = \sigma^2_{\text{prior}} \left( 1 - \frac{\sigma^2_{\text{prior}} e^2}{1 + \sigma^2_{\text{prior}} e^2} \right) = \frac{\sigma^2_{\text{prior}}}{1 + \sigma^2_{\text{prior}} e^2}. \tag{51}$$

## 9.4 Efficient implementation of BAdam

See Alg. 2 for a complete specification of our approach. There are two important points to note. First, we have carefully ordered the updates so that they can all be done in-place, improving memory efficiency. Second, we have simplified the derivation, by defining,

$$\boldsymbol{T} = \boldsymbol{I} - \boldsymbol{A}. \tag{52}$$

---

**Algorithm 1** BRMSprop

| | |
|---|---|
| **Require:** $\eta$ | $\triangleright$ Learning rate |
| **Require:** $\eta_{\mathrm{g}}$ | $\triangleright$ Learning rate for average gradient (0.01) |
| **Require:** $\sigma^2$ | $\triangleright$ Prior variance ($1/(2N_{\mathrm{inputs}})$) |
| 1: $\Sigma \leftarrow \sigma^2$ | $\triangleright$ Initialize uncertainties |
| 2: $\mu \sim \mathcal{N}\left(0, 2/N_{\mathrm{inputs}}\right)$ | $\triangleright$ Initialize parameters |
| 3: $\hat{g} \leftarrow 0$ | $\triangleright$ Initialize average gradient |
| 4: **while** not converged **do** | $\triangleright \Sigma = \sigma_{\mathrm{prior}}^2(t),\, \mu = \mu_{\mathrm{prior}}(t)$ |
| 5: $\quad g \leftarrow \nabla \mathcal{L}_t(\mu)$ | $\triangleright$ Gradients of minibatch $t$ |
| 6: $\quad \Sigma \leftarrow \Sigma/(1 + (g - \hat{g})^2 \Sigma)$ | $\triangleright \Sigma = \sigma_{\mathrm{post}}^2(t)$ |
| 7: $\quad \mu \leftarrow \mu + \Sigma g$ | $\triangleright \mu = \mu_{\mathrm{post}}(t)$ |
| 8: $\quad \hat{g} \leftarrow (1 - \eta_{\mathrm{g}})\,\hat{g} + \eta_{\mathrm{g}} g$ | $\triangleright$ Update average gradient |
| 9: $\quad \Sigma \leftarrow \left(1 - \eta^2/(2\sigma^2)\right)^2 \Sigma + \eta^2$ | $\triangleright \Sigma = \sigma_{\mathrm{prior}}^2(t+1)$ |
| 10: $\quad \mu \leftarrow \left(1 - \eta^2/(2\sigma^2)\right)\,\mu$ | $\triangleright \mu = \mu_{\mathrm{prior}}(t+1)$ |
| 11: **end while** | |
| 12: **return** $\mu$ | |

---

## 9.5 TRICK TO ALLOW LEARNING RATES TO VARY ACROSS PARAMETERS

We require that the initialization for $\Sigma$ varies across parameters, according to the number of inputs (as in typical neural network initialization schemes). While this is possible in automatic differentiation libraries, including PyTorch, it is extremely awkward. As such, we reparameterise the network, such that all parameters are initialized with a draw from a standard Gaussian, and to ensure that the outputs have the same scale, we explicitly scale the output.

## 9.6 NESTEROV ACCELERATED GRADIENT

To begin, we note that accelerated stochastic gradient descent remains an open research problem, in which the optimality of plain NAG is unclear (Cohen et al., 2018). As such, our goal is again, not to match NAG exactly, but to capture its essential insights within a Bayesian framework, so that we can suggest improved methods. To highlight the link between our approach and Nesterov accelerated gradient, we rewrite Eq. (32) in terms of $v(t+1) = \eta \mu_{\mathrm{p}}(t+1)$,

$$\mu_{\mathrm{w}}(t) \approx \mu_{\mathrm{w}}(t-1) + v(t) \tag{53a}$$

$$v(t) \approx (1 - \eta_{\mathrm{p}})\, v(t-1) + \eta \eta_{\mathrm{p}} \frac{g(t)}{\sqrt{\langle e^2 \rangle}}, \tag{53b}$$

which results in updates with a very similar form to standard momentum and Nesterov accelerated gradient (e.g. Sutskever et al., 2013), with the addition of root-mean-square normalization for the gradient. The key difference between momentum and Nesterov accelerated gradient is where we evaluate the gradient: for momentum, we evaluate the gradient at $\mu_{\mathrm{w}}(t-1)$ (i.e. at $\mu_{\mathrm{post}}(t-1)$), and for Nesterov accelerated gradient, we evaluate the gradient at a "predicted" location (i.e. at $\mu_{\mathrm{prior}}(t)$),

$$g_{\mathrm{Mom}}(t) = g\left(\mu_{\mathrm{w}}(t-1)\right) = g\left(\mu_{\mathrm{post, w}}(t-1)\right) \tag{54a}$$

$$g_{\mathrm{NAG}}(t) = g\left(\mu_{\mathrm{w}}(t-1) + (1 - \eta_{\mathrm{p}})v(t-1)\right) \approx g\left(\mu_{\mathrm{w}}(t-1) + v(t-1)\right) = g\left(\mu_{\mathrm{prior, w}}(t)\right) \tag{54b}$$

As noted in Eq. (17), Bayesian filtering requires us to evaluate the gradient at $\mu_{\mathrm{prior}}$, implying that we use updates based on NAG (Eq. 54b), rather than updates based on standard momentum (Eq. 54a).

## 9.7 ESTIMATING THE NEGATIVE HESSIAN

One approach that works well in practice in natural-gradient methods (Amari, 1998; Martens & Grosse, 2015; Grosse & Martens, 2016; Song & Ermon, 2018; Smith et al., 2018) is to use a Fisher-

---

**Algorithm 2** BAdam

---

**Require:** $\eta$                                                                                                ▷ Learning rate
**Require:** $\eta_g$                                                              ▷ Learning rate for average gradient (0.01)
**Require:** $\eta_p$                                                                      ▷ Learning rate for momentum (0.1)
**Require:** $\eta_w$                                                                       ▷ Learning rate for weight alone (0)
**Require:** $\sigma^2$                                                                            ▷ Prior variance $(1/(2N_{inputs}))$
  1: $T_{ww} = 1 - (\eta^2 + \eta_w^2)/(2\sigma^2)$                                           ▷ Set dynamics matrix
  2: $T_{wp} = \eta$                                                                       ▷ Set dynamics matrix
  3: $T_{pp} = 1 - \eta_p$                                                                    ▷ Set dynamics matrix
  4:
  5: $\Sigma_{ww} \leftarrow \sigma^2$                                                            ▷ Initialize uncertainties
  6: $\Sigma_{wp} \leftarrow 0$                                                                  ▷ Initialize uncertainties
  7: $\Sigma_{pp} \leftarrow 0$                                                                  ▷ Initialize uncertainties
  8: $\mu_w \sim \mathcal{N}\left(0, 2/N_{inputs}\right)$                                              ▷ Initialize parameter
  9: $\mu_p \leftarrow 0$                                                                     ▷ Initialize momentum
 10: $\hat{g} \leftarrow 0$                                                                   ▷ Initialize average gradient
 11: **while** not converged **do**
 12:     $g \leftarrow \nabla \mathcal{L}_t(\mu_w)$                                                    ▷ Gradients of minibatch $t$
 13:     $e^2 \leftarrow \nabla(g - \hat{g})^2$
 14:     $\Sigma_{ww} \leftarrow \Sigma_{ww} - \Sigma_{ww}^2 e^2/(1 + \Sigma_{ww}e^2)$
 15:     $\Sigma_{wp} \leftarrow \Sigma_{wp} - \Sigma_{ww}\Sigma_{wp}e^2/(1 + \Sigma_{ww}e^2)$                       ▷ $\boldsymbol{\Sigma} = \boldsymbol{\Sigma}_{post}(t)$
 16:     $\Sigma_{pp} \leftarrow \Sigma_{pp} - \Sigma_{wp}^2 e^2/(1 + \Sigma_{ww}e^2)$
 17:     $\mu_w \leftarrow \mu_w + \Sigma_{ww}g$                                                     ▷ $\boldsymbol{\mu} = \boldsymbol{\mu}_{post}(t)$
 18:     $\mu_p \leftarrow \mu_p + \Sigma_{wp}g$
 19:     $\hat{g} \leftarrow (1 - \eta_g)\hat{g} + \eta_g g$                                               ▷ Update average gradient
 20:     $\Sigma_{ww} \leftarrow T_{ww}^2\Sigma_{ww} + 2T_{ww}T_{wp}\Sigma_{wp} + T_{wp}^2\Sigma_{pp} + \eta_w^2$
 21:     $\Sigma_{wp} \leftarrow T_{ww}T_{pp}\Sigma_{wp} + T_{wp}T_{pp}\Sigma_{pp}$                            ▷ $\boldsymbol{\Sigma} = \boldsymbol{\Sigma}_{prior}(t+1)$
 22:     $\Sigma_{pp} \leftarrow T_{pp}^2\Sigma_{pp} + \eta_p^2$
 23:     $\mu_w \leftarrow T_{ww}\mu_w + T_{wp}\mu_p$                                                 ▷ $\boldsymbol{\mu} = \boldsymbol{\mu}_{prior}(t+1)$
 24:     $\mu_p \leftarrow T_{pp}\mu_p$
 25: **end while**
 26: **return** $\mu$

---

Information based estimate, of the negative Hessian,

$$\Lambda_{like} \approx E_{P(d_{model}|\boldsymbol{z})}\left[-\frac{\partial^2}{\partial \boldsymbol{z}^2}\log P\left(d_{model}|\boldsymbol{z}\right)\right] = E_{P(d_{model}|\boldsymbol{z})}\left[\boldsymbol{e}\boldsymbol{e}^T\right], \tag{55}$$

where

$$\boldsymbol{e} = \left.\frac{\partial \log P\left(d_{model}|\boldsymbol{z}\right)}{\partial \boldsymbol{z}}\right|_{\boldsymbol{z}=\boldsymbol{\mu}_{prior}}. \tag{56}$$

Importantly, this Fisher-Information based estimate does not depend on actual data, $d$. Instead, it uses expectations (and gradients) taken over data sampled from the model, $d_{model}$, and this is fundamentally necessary for the the second equality in Eq. (55) to hold. As such, to use Eq. (55) we need to sample surrogate data from the model, and compute gradients with respect to that sampled data. Unfortunately, sampling surrogate data increases complexity and the required computational cost. As such, a simple low-cost alternative is to note that the key difference between $\boldsymbol{e}$ and $\boldsymbol{g}$ is that $\boldsymbol{e}$ has zero-mean, whereas $\boldsymbol{g}$ has non-zero mean, suggesting that we should correct the gradient by subtracting its mean,

$$\boldsymbol{e} \approx \boldsymbol{g} - \hat{\boldsymbol{g}}$$

where $\hat{\boldsymbol{g}}$ is an empirical estimate of the mean gradient.

