# OpenReview forum: "A unified theory of adaptive stochastic gradient descent as Bayesian filtering"
_ICLR.cc/2019/Conference_

### Official Review · AnonReviewer3 · 2018-11-02
**Unnatural approximations**

**Rating:** 5
**Confidence:** 4

**Review:**

In this work, the authors attempt to unify existing adaptive gradient methods under the Bayesian filtering framework with the dynamical prior.  In Ollivier, 2017, a framework is proposed to connect Bayesian filtering and natural gradient.  On the other hand,  in Khan et al., 2018. an approach is proposed to connect natural gradient and adaptive gradient methods.  The main contributions of this work are (1)  introducing a dynamical prior and (2) recovering RMSProp and Adam as special cases.

However, the proposed dynamical prior is very similar to the fading memory technique used in Ollivier, 2017. (see Proposition 3 of Ollivier, 2017)
Furthermore, the authors argue that this work recovers a root-mean-square form while Khan et al., 2018 recovers a different sum-square form. Unfortunately, the authors have to use a series of unnatural approximations to recover the root-mean-square form. In fact, as mentioned in Khan, 2017b  this proposed method without these approximations is also a mean-square form. (also see Eq (2.28-2.29) of Ollivier, 2017)

Since the authors mainly follow Ollivier, 2017 and make unnatural approximations,  the work has a limited impact.  To get a higher rating, the authors should clearly give justifications and insights of these approximations.

Detailed comments:
(1) On Page 1,  "The typical approach to Bayesian filtering, where we infer a distribution, ... jointly, forces us to use extremely strong, factorised approximations, and it is legitimate to worry that these strong approximations might meaningfully disrupt the ability of Bayesian filtering to give close-to-optimal updates.   ... we instead consider ... that incorporates factorisation into the problem setting, and therefore requires fewer approximations downstream. "
The proposed method is equivalent to jointly perform Kalman filtering with full-covariance with an additional diagonal-approximation step. This additional step might also meaningfully disrupt the ability of Bayesian filtering. Furthermore, such approximation ignores the off-diagonal terms in the low-rank approximation at Eq (8).

Minor: You should use \approx at Eq (8) since a rank-1 approximation is used.

(2) On page 2, "It has been noted that under specific circumstances, natural gradient is approximate Bayesian filtering (Ollivier, 2017), allowing us to link Bayesian filtering to the rich literature on natural gradients.  However, this only occurs when the dynamical prior in the Bayesian filtering problem has a specific form: the parameters being fixed over time (i.e.  arguably an online data, rather than a true Bayesian filtering setting)."
The authors should comment the difference between the dynamical prior and the fading memory technique (see Proposition 3 of Ollivier, 2017) where at page 14 of Ollivier, 2017, Ollivier mentions that "this is equivalent ... or to the addition of an artificial process noise ... in the model".  I think Ollivier's idea is very similar to the dynamical prior used at Eq (1) of this submission.  Furthermore, the second-order Taylor expansion with a Fisher information-based estimation of Hessian (see the equation below Eq(1) of this submission) is exactly the same as Ollivier's Extended Kalman filter (see  Eq 2.25 at Lemma 9  and Lemma 10 of Ollivier, 2017).  The authors should cite Ollivier, 2017.

Minor: Eq (6) should be E_p [ - \nabla_z^2 \log p(d|z) ] = E_p  [ e e^T ], where "-", the negative sign is missing. Please see the definition of the Fisher information matrix.

(3) On page 2, "While there have been attempts to use natural gradients to recover the Adam or RMSprop root-mean-square form for the gradient normalizer, in practice a different sum-square form emerges (Khan & Lin, 2017; Khan et al., 2018). In contrast, we show that to recover the Adam or RMSprop form for the gradient normalizer."
Khan et al., 2018 is a mean-square form for variational inference due to the entropy term of the variational distribution. (see Sec 3 and 5 of  Khan et al., 2018 and Khan, 2017b )
Unfortunately, the "root-mean-square form" does not appear naturally in this submission. In practice, the proposed update is also a mean-square form  (see Eq (2.28-2.29) of Ollivier, 2017 and Khan, 2017b) without a series of unnatural approximations used in this submission.
To justify these assumptions, the authors should explain when "the steady state posterior variance" (see sec 2.21) and  "a self-consistent solution" (see sec 7.1) achieve.  As far as I know, \sigma^2_t = \sigma^2_{t+1} in sec 2.2.1 only holds in the limit case when t-> \inifity.  Why does the equality hold at each time step t? The authors should give a justification or an intuition about these approximations since this paper is a theory paper. Please also see my next point.

(4) Section 7.1 is also confusing.
In sec 7.1, the authors assume that A \in O(\eta). However, A=\eta^2/(2\sigma^2) in sec 2.2 and A_{1,1} =  ( \eta_w^2+\eta^2 )/ (2\sigma^2) at Eq (14). In both cases, A can be \in O(\eta^2). This is very *critical* since the authors argue that O(\eta^3) can be neglected in sec 7.1.  The authors use this point to show that Adam is a special case.
If A \in O(\eta^2), we know that "A \Sigma_{post}" \in O(\eta^3) should be neglected. At the last equation on page 10,  the authors do not neglect "A \Sigma_{post}". Why?  The authors should clarify this point to avoid doing *selective* neglection.  Again, the impact of this paper should be inspiring new adaptive methods.
The authors also mention that the second-order term in A is neglected in sec 7.2. Any justification?


References
[1] Ollivier, Yann. "Online Natural Gradient as a Kalman Filter." arXiv preprint arXiv:1703.00209 (2017).
[2] Khan, Mohammad Emtiyaz, and Wu Lin. "Conjugate-computation variational inference: Converting variational inference in non-conjugate models to inferences in conjugate models." arXiv preprint arXiv:1703.04265 (2017).
[3] Khan, Mohammad Emtiyaz, et al. "Vprop: Variational Inference using RMSprop." arXiv preprint arXiv:1712.01038 (2017b).
[4] Khan, Mohammad Emtiyaz, et al. "Fast and Scalable Bayesian Deep Learning by Weight-Perturbation in Adam" (2018)

---

> ### Author Response · Authors · 2018-11-15
> **Response (2/2)**
>
> (2) We have included a paragraph in the introduction on the difference with Olliver's (2017) fading memory technique.  To reiterate, they use a highly unnatural generative process under which the process noise depends on inferences under that generative model (in particular, the posterior covariance).  In contrast, the process noise under our model is "meaningful" in the sense that it does not depend on inferences under the model.
>
> Minor: we have delete this equation.
>
> (3) Steady-state is indeed only reached as t -> infinity, and we have added a note to this effect.  At the very least, we don't expect to obtain a mean-square form for the normalizer, because that would require us to follow Olliver (2017) in using process noise proportional to our uncertainty, which we categorically do not do.
>
> (4) The first point is that we agree, the impact of this paper should be inspiring new adaptive methods.  And we do just that, with BRMSprop and BAdam.  Importantly, these approximations are never used in simulations/updates for BRMSprop and BAdam.  Instead, they are only used to think about the similarities and differences between our Bayesian approaches (BRMSprop and BAdam) and classical methods such as RMSprop or Adam.  As such, in some sense, the stronger the approximations we need to make BAdam close to Adam, the more scope we have for developing improved adaptive methods!
>
> In response to your specific questions.
> Here, we are considering the limit of \eta -> 0, so assuming A scales with \eta is weaker than assuming A scales with \eta^2.  While A for RMSprop indeed scales with \eta^2, A for Adam scales with \eta (the top-right element is -eta). As A in general scales with \eta, the terms you refer to, A S_post, scale with \eta^2, and cannot be neglected in general.  However, for the case of RMSprop, A indeed scales with eta^2, and so we can neglect these terms, and indeed we do just that in the original draft (the approximate variance updates that we solve for in steady state do not include any weight-decay terms).  We have clarified this point in the supplementary, including both the RMSprop and Adam cases.
>
> As a technical issue, the usual definition of big-O notation considers the limit as \eta -> \infty, whereas the relevant limit in our case is \eta -> 0.  I have therefore replaced the \in big-O notation, with the physics-inspired \sim (for "scales with").

---

> > ### Comment · AnonReviewer3 · 2018-11-15
> > **incorrect statements**
> >
> >  I  list some incorrect statements made in this revised version.
> >
> >
> > (1) "Moreover, neither approach gives rise to momentum."
> > This statement is not true.
> > The authors should read Sec 4 of [4].
> >
> > (2) "Notably, 2 is very, very artificial: I have never seen a Bayesian generative model which depends on inferences under that model.  Our approach does not use this process noise, and as such, if nothing else, our generative process is more meaningful, in the very basic sense that it doesn't depend on the posterior."
> > "Thus, their generative model depends on inferences made under that model: a highly unnatural assumption that most likely does not correspond to any “real” generative process."
> >
> > Note that the fading memory technique and the extended Kalman filter both depend on inferences under that model. (please read [5])
> > As far as I know, the proposed method is an extended Kalman filter, which also depends on inferences under that model.
> > I strongly do not like this statement since the authors make unnatural approximations hidden in the derivation while the authors claim their method is more natural than [1].
> >
> > (3) "the Hessian is always rank 1"
> > I do not think this statement is correct. The main reason to use g g^T (or e e^T) is due to the empirical Fisher estimation, which implies that the true Hessian is not a rank 1 matrix.
> >
> >
> > References:
> > [1] Ollivier, Yann. "Online Natural Gradient as a Kalman Filter." arXiv preprint arXiv:1703.00209 (2017).
> > [4] Khan, Mohammad Emtiyaz, et al. "Fast and Scalable Bayesian Deep Learning by Weight-Perturbation in Adam" (2018)
> > [5] Humpherys, Jeffrey, Preston Redd, and Jeremy West. "A fresh look at the Kalman filter." SIAM review 54.4 (2012): 801-823.

---

> > > ### Author Response · Authors · 2018-11-15
> > > **Response**
> > >
> > > (1) Presumably, you mean AR(2)?  Vanilla gradient descent has AR(1) like dynamics, as the next position depends on the previous one, whereas momentum-based methods can be formulated as depending on the past two positions, where we estimate the velocity using their difference.
> > >
> > > Regarding [1], we can always take an inference problem, convert it to an optimisation problem using VI, and apply momentum to perform the optimisation.  This is always possible, and is always a valid approach.  But this is using, rather than recovering momentum.  In contrast, in our work, we are interested in how momentum might emerge as the solution to a principled inference problem.
> > >
> > > An alternative viewpoint is to think about your reference (Olliver 2017).  As long as they're doing natural gradient, the link to Bayesian filtering and hence to Bayesian inference is retained. As soon as they stop doing natural gradient (e.g. by adding momentum) the link to Bayesian filtering is broken, and they are simply using some optimisation strategy (though perhaps a very effective one) to optimize a loss function (albeit one that happens to be motivated by Bayesian considerations).
> > >
> > > (2) Under the standard set-up for an EKF, the generative model does not depend on inferences made under that model.  For instance, one might sample the latent variable z from the following process,
> > > z(t+1) ~ N((I - A) z(t), Q),
> > > and the observed data, g, from another Gaussian,
> > > g(t+1) ~ N(f(z(t)), I)
> > > Note that as we have an arbitrary function, f(z(t)), here, we are in the EKF setting that you describe.
> > >
> > > To reiterate, this is the true, underlying generative model, and it never depends on inferences made under the model.  To perform extended Kalman filtering, we can define an approximate model as a linearisation of f around the current mean.  However, that is categorically not what is going on in Olliver (2017).  There they set the process noise, Q, equal to their filtering uncertainty.  Olliver (2017) do not claim, and it is not, to my knowledge possible to set up a valid generative model (i.e. one for which the process noise, Q, is a function of z) under which this emerges in an EKF-like fashion.
> > >
> > > Please let me know if there are any other issues I can clarify.  I await with interest your opinion on our new derivations that treat the back propagated gradient as the data in a Bayesian inference problem!

---

> > > > ### Comment · AnonReviewer3 · 2018-11-15
> > > > **issue (2)**
> > > >
> > > > "  Olliver (2017) do not claim, and it is not, to my knowledge possible to set up a valid generative model (i.e. one for which the process noise, Q, is a function of z) under which this emerges in an EKF-like fashion"
> > > >
> > > > On Page 10 of  Ollivier 2017, " (Note, however, that the online natural gradient and extended Kalman filter are identical at every time step, not only asymptotically.)"  Also see Definition 5 on page 16 of  Ollivier 2017.
> > > >
> > > > To reiterate, EKF itself depends on inferences under that model due to the Gaussian approximation.

---

> > > > > ### Author Response · Authors · 2018-11-15
> > > > > **Response**
> > > > >
> > > > > To reiterate.  The EKF is an inference technique.  The underlying prior (just like all other prior distributions in the Bayesian literature) does not depend on inferences under that model.  For instance, on the Wikipedia page for the EKF (https://en.wikipedia.org/wiki/Extended_Kalman_filter), under "formulation", they define the prior, which makes reference to nonlinear functions, but not to inferences made under the model.
> > > > >
> > > > > Of course, when performing inference, ("Discrete-time predict and update equations" on Wikipedia), they do make reference to the inferred means and uncertainties.  To perform inference (and only to perform inference), they linearise the nonlinear functions, around the current inferred mean.  To reiterate, there is --- and has to be --- an underlying nonlinear model that does not depend on the inferences.  Only the surrogate, approximate linearised model depends on the inferences, and then, only through the location at which you perform the Taylor expansion.
> > > > >
> > > > > I agree Olliver 2017 obtain Kalman Filter updates that are equivalent to natural gradient at every step.  However, this requires them to use process noise, Q, proportional to the filtering covariance, P, i.e. they use Q = Q(P).  It is not possible, to my knowledge, to define a nonlinear EKF style model as in (https://en.wikipedia.org/wiki/Extended_Kalman_filter), under "formulation" which displays the same behaviour i.e. Q(P) = Q(z), and, again to reiterate, Olliver 2017 do not claim as such.
> > > > >
> > > > > That said, for the main thrust of our argument, the artificiality or otherwise for the Olliver 2017 process noise is a side point.  Olliver 2017 work hard to ensure that their approach is equivalent to natural gradient, but that means they can't recover momentum, and don't recover the root-mean-square form for the gradient normaliser.  In contrast, we give a principled justification for a different Kalman filtering approach (with a different choice for the process noise) that is able to recover momentum, and the root-mean-square gradient normaliser.

---

> > > > > > ### Comment · AnonReviewer3 · 2018-11-15
> > > > > > **no**
> > > > > >
> > > > > > (1) "The underlying prior (just like all other prior distributions in the Bayesian literature) does not depend on inferences under that model.  For instance, on the Wikipedia page for the EKF (https://en.wikipedia.org/wiki/Extended_Kalman_filter), under "formulation", they define the prior, which makes reference to nonlinear functions, but not to inferences made under the model."
> > > > > >
> > > > > > Note that the EKF linearizes the non-conjugate/non-Gaussian likelihood or the observation likelihood. For example, the observation can be a positive definitive matrix. In this case, a Wishart likelihood should be used.  In Ollivier 2017, this approximation is clearly stated. Needless to say, the likelihood depends on the model.
> > > > > >
> > > > > > (2) "In contrast, we give a principled justification for a different Kalman filtering approach (with a different choice for the process noise) that is able to recover momentum, and the root-mean-square gradient normalizer."
> > > > > >
> > > > > > I think the authors should define "a principled justification."
> > > > > > If my understanding is correct, a principled justification should come without unjustified approximation. To recover the root-mean-square gradient normalizer, additional approximations have to be made.  Do the authors justify these approximations? Why these approximations are natural.
> > > > > > My concern is that whether these approximations come from the first principle.  Honestly, it feels contrived in order to recover RmsProp and Adam, which implies that the paper has a limited impact. To address this, the authors should show that non-diagonal adaptive gradient methods can be derived naturally without case-by-case approximations. For example, the authors should give a non-diagonal version of (Bayesian) RmsProp or Adam using the *exactly* same approximations made in the paper.

---

> > > > > > > ### Author Response · Authors · 2018-11-16
> > > > > > > **Response**
> > > > > > >
> > > > > > > (1) Indeed, the EKF defines an approximate linearised likelihood, as a Taylor series expansion of the underlying nonlinear model.  The underlying non-linear model cannot, and should not depend on inferences made under the model.  Only the linearised model depends on the inferred mean for the latent variable, and then, only through the location at which you perform the linearisation.
> > > > > > >
> > > > > > > We entirely agree that the approximate, linearised likelihood does indeed depend on inferences made under the model, and that this a perfectly natural, sensible thing to do.  This is not the question.  The question is whether the dynamical prior (in particular the process noise) depends on inferences made under the model.  Olliver 2017 set the process noise proportional to the posterior uncertainty (covariance), and it is this that can't be written down within the EKF framework.
> > > > > > >
> > > > > > > (2) I would kindly request that you read the new section, entitled "Factorisation implies a rich dynamical prior", which provides the principled justification that you request.
> > > > > > >
> > > > > > > Again, it isn't really an approximation, but a new way of setting up the problem.  In particular, we note that the updates in most algorithms for neural network optimization are based on just the gradient for that parameter.  As such, if we define a factorised model for each parameter separately, we can take the "data" to be not the underlying input-output pairs, but the backpropagated gradient for that parameter.  This generative model naturally gives rise to the emergence of a new latent variable, w_i^*, the optimal value for the ith parameter, conditioned on the current estimate of all the other parameters.  If we choose to work with this parameter, then dynamics emerge automatically: w_i^* must change over time, because it depends on our current estimate of all the other parameters, which are changing as they are optimized.
> > > > > > >
> > > > > > > The exact same exercise can trivially be extended to non-diagonal approximations using the derivations in our paper ("Bayesian (Kalman) filtering as adaptive SGD"), and we are considering the Kronecker factored approach in future work.  However, we believe that the we have already made considerable contributions: an entirely novel connection between high-dimensional correlations and temporal changes in a principled Bayesian model, that recovers state-of-the-art adaptive methods including RMSprop, Adam, AdamW and NAG.

---

> > > > > > > > ### Comment · AnonReviewer3 · 2018-11-16
> > > > > > > > **Review of the revised version**
> > > > > > > >
> > > > > > > >
> > > > > > > > Since this paper is a theory paper, my evaluation criterion is based on the mathematical justification.
> > > > > > > > As claimed, the main contribution of this paper is (1) introducing the dynamical prior and (2) recovering a root-mean-square normalizer such as (non-Bayesian) RMSProp/Adam as a special case.
> > > > > > > >
> > > > > > > > Although the authors claim their framework is more natural than [1],
> > > > > > > > the main concern is the underlying unnatural approximations made in the derivation. As suggested by [1], the Bayesian RmsProp and Bayesian Adam indeed admit a mean-square normalizer without further approximations.   To recover (non-Bayesian) RmsProp/Adam, which is a root-mean-square normalizer, the authors have to make additional approximations to get the exact RmsProp/Adam update. I do not think these additional approximations are natural.
> > > > > > > >
> > > > > > > > Detailed Issues
> > > > > > > > (1) On page 1, "their generative model depends on inferences
> > > > > > > > made under that model: a highly unnatural assumption that most likely does not correspond to any “real” generative process."
> > > > > > > > The paper and [1] use the extended Kalman filter to linearize the non-Gaussian likelihood.
> > > > > > > > Extended Kalman filter and fading-memory technique are known as standard estimation techniques. (see [5] and [6]). The fading-memory technique can be viewed as "an artificial process noise proportional to [the posterior covariance" as mentioned at [1]. To be fair, I do not think the fading memory technique is a highly unnatural assumption given that non-standard and unjustified approximations are made later in paper to recover the (non-Bayesian) RmsProp/Adam.
> > > > > > > >
> > > > > > > > (2) I do not think that the equivalence between Eq (1) and Eq (2) is obvious. A proof is required.
> > > > > > > >
> > > > > > > > (3) In figure 1, in order to get a traceable algorithm, an additional approximation is made. (figure 1.C is an approximation of figure 1.B)
> > > > > > > > Is this kind of approximation an estimation technique? Why this is better/(more natural) than the fading memory technique?
> > > > > > > >
> > > > > > > > (4) If A ~ \eta and Q ~ \eta^2 and O(\eta^3) terms are ignored, I do not think  the first equation below Sec 9.1 can exactly recover the followling equation on page 12.
> > > > > > > > \Sigma_post^(t + 1) =\Sigma_post − A \Sigma_post − \Sigma_post A^T + Q − \Sigma_post ee^T  \Sigma_post
> > > > > > > >
> > > > > > > > (5) " Neglecting the second-order term in A" from the line above Eq 34 on page 13
> > > > > > > > To obtain Eq 34, the authors have to ignore  "A \Psi A^T". Why do the authors neglect "A \Psi A^T" in the derivation?  Any justification?
> > > > > > > >
> > > > > > > > (6) Finally, with approximations made in (4) and (5), (non-Bayesian) RmpProp/ Adam is recovered in the limit case when t -> \infinity. (see sec 4.1 and 5.1). The proposed method cannot recover (non-Bayesian) RmsProp/Adam or a root-mean-square normalizer at an early stage since t is not large enough.
> > > > > > > >
> > > > > > > > I may add more approximation issues made in Sec 2.
> > > > > > > >
> > > > > > > > References
> > > > > > > > [1] Ollivier, Yann. "Online Natural Gradient as a Kalman Filter." arXiv preprint arXiv:1703.00209 (2017).
> > > > > > > > [5] Humpherys, Jeffrey, Preston Redd, and Jeremy West. "A fresh look at the Kalman filter." SIAM review 54.4 (2012): 801-823.
> > > > > > > > [6] Musoff, Howard, and Paul Zarchan. Fundamentals of Kalman filtering: a practical approach. American Institute of Aeronautics and Astronautics, 2009.

---

> > > > > > > > > ### Author Response · Authors · 2018-11-17
> > > > > > > > > **Response**
> > > > > > > > >
> > > > > > > > > (1) Indeed, the fading-memory technique can be viewed as "an artificial process noise proportional to [the posterior covariance]".  We use a different process noise that is more natural (in the sense that the process noise is fixed, rather than depending on inferences made under the model).  Further, this different process noise gives a very different, behaviour of the update rules from the mean-square normaliser that requires this form for the process noise.  I should also note that all of our approximations are very standard and well-understood in the Physics and Engineering literature, to the extent that they are arguably high school, rather than undergraduate level (e.g. steady state, neglecting terms such as A Phi A proportional to dt^2).
> > > > > > > > >
> > > > > > > > > (2) This is obvious, because for a well-calibrated model, the posterior is well-calibrated, and thus the true parameter can be understood as being drawn from the posterior.  Further, in (2), we take the expectation under the true data distribution --- in effect, giving us infinite data.  Of course, in the infinite data regime, we again recover the underlying parameters.
> > > > > > > > >
> > > > > > > > > To confirm the validity of the approximations in this section, we show that the second-order approximation to the likelihood is the same under the model conditioned on the gradient, and under the original model.
> > > > > > > > >
> > > > > > > > > (3) It is well known that an autoregressive process of any order can be written as an AR(1) process by defining an expanded state space.  See Example 1 in lecture notes: http://faculty.washington.edu/ezivot/econ584/notes/statespacemodels.pdf
> > > > > > > > > In that link, they go so far as regarding this equivalence as "obvious".
> > > > > > > > >
> > > > > > > > > (4) Could you be more specific?  What do you get?  If you're worried about the denominator, that term is basically,
> > > > > > > > >
> > > > > > > > > Sigma e e^T Sigma / (1 + e^T Sigma e)
> > > > > > > > >
> > > > > > > > > Taking a first-order Taylor series expansion of the denominator, we obtain,
> > > > > > > > >
> > > > > > > > > Sigma e e^T Sigma (1 - e^T Sigma e)
> > > > > > > > >
> > > > > > > > > But as Sigma ~ eta, and we neglect terms that scale with eta^3, we have, this reduces to just Sigma e e^T Sigma
> > > > > > > > >
> > > > > > > > > (5) This is a standard approximation made in almost all work on dynamical systems in engineering and physics.  Specifically, it is valid because Phi~1, and
> > > > > > > > > for RMSprop:
> > > > > > > > > A~eta^2
> > > > > > > > > so A Phi A ~ eta^4 is dominated by A Phi ~ eta^2, and can be neglected
> > > > > > > > > and for Adam:
> > > > > > > > > A~eta
> > > > > > > > > so A Phi A~ eta^2 is dominated by A Phi ~ eta, and can be neglected.  Note that, phrased in this way, this constitutes taking the "leading order" terms in an expression, and is again a standard, well-understood approximation.
> > > > > > > > >
> > > > > > > > > (6) Indeed, there is a difference at small t, and this may be at the root of our improved performance.

---

> > > > > > > > > > ### Comment · AnonReviewer3 · 2018-11-17
> > > > > > > > > > **Ad hoc approximations**
> > > > > > > > > >
> > > > > > > > > > (1) "I should also note that all of our approximations are very standard and well-understood in the Physics and Engineering literature, to the extent that they are arguably high school, rather than undergraduate level (e.g. steady state, neglecting terms such as A Phi A proportional to dt^2). "
> > > > > > > > > >
> > > > > > > > > > I think in the Physics and Engineering literature, approximation conditions are well-defined and studied.  If many ad-hoc approximations are introduced, do these approximation conditions jointly hold in this case?
> > > > > > > > > >
> > > > > > > > > > For example,in Sec 9.1, the authors argue that "to obtain a self-consistent solution, we need, \Sigma_{post} ~ O(\eta)".
> > > > > > > > > >  Why does this assumption hold? Why the self-consistent solution implies that \Sigma_{post} ~ O(\eta)? A proof is required.  Does the update (such as Eq 15) meet this assumption?  As t -> \initify, can the authors show that  "\Sigma_{post} ~ O(\eta)"?
> > > > > > > > > > In Sec 9.1, the authors further use "a first-order Taylor series expansion of the denominator" while some terms are neglected in the numerator. Is it a standard, well-understood approximation? Note that Sec 9.1 is critical since the authors use the results in Sec 9.1 to recover non-Bayesian RmsProp and Adam.
> > > > > > > > > >
> > > > > > > > > > To reiterate, do the authors consider the *joint* side effect of these local approximations hidden in the whole derivation? Note that the approximation error can be cumulative. The dynamical system could be unstable due to the cumulative error.
> > > > > > > > > >
> > > > > > > > > > Every time the authors make an approximation, they should consider the side effect of the approximation. If they make many approximations, they should also consider the interactions between these approximations and the cumulative approximation error. The authors can cite existing works to justify the approximations as long as existing works use the same approximations. Please remember this is a *theory* paper.
> > > > > > > > > >  A theory paper should have formal proofs to justify all approximations made in the paper.  There are ad hoc approximations hidden in the derivation.  Also, the empirical results are weak, which is fine since this is a theory paper. Moreover, if the authors make assumptions, these assumptions should be clearly mentioned before going to the derivation/proof. Unfortunately, many approximations and assumptions are hidden in the derivation in this paper. I do not think this paper should be considered as a theory paper. I suggest that the authors should re-frame this paper as an algorithm paper, propose a non-diagonal version of adaptive gradient methods, and empirically examine the approximations made in this paper.
> > > > > > > > > >
> > > > > > > > > >
> > > > > > > > > > (2) "because for a well-calibrated model" "in the infinite data regime" A big IF. The argument is very hand-waving.  I do not think that the argument is formal.   In Eq(1), w^* is drawn from a posterior while in Eq (2), w^* is the argmax of the expectation. Why is the equivalence obvious? A proof is required. The authors can cite an existing work.
> > > > > > > > > >
> > > > > > > > > > (3) It is an approximation. Is it correct? As mentioned at http://faculty.washington.edu/ezivot/econ584/notes/statespacemodels.pdf, to recover an autoregressive process of any order, the state space must be augmented. For a full order with infinite time steps as suggested in Figure 1.A, the dimensionality of the augmented state space will be infinite.  Note that in the stochastic gradient setting, the original state space is the parameter space. If Figure 1.C is exactly equivalent to Figure 1.B, the dimensionality of z should be |T| \times dim(w), where |T| is the number of time steps and dim(w) is the dimensionality of the parameter space. However, in Eq 10,  dim(z_{rmsprop}) = dim(w). It is clear that the authors use the first-order approximation in the RmsProp case, where Figure 1.C is an approximation of Figure 1.B.
> > > > > > > > > >
> > > > > > > > > >
> > > > > > > > > > (4)  " for Adam: A~eta
> > > > > > > > > > so A Phi A~ eta^2 is dominated by A Phi ~ eta, and can be neglected. "
> > > > > > > > > >
> > > > > > > > > > In Sec 9.1,  the authors argue that "As \Sigma_post is O(\eta), and updates to \Sigma_post are O(\eta^2), we can neglect O(\eta^3) terms"
> > > > > > > > > >
> > > > > > > > > > Why not ignore updates to \Sigma_post  since "updates to \Sigma_post are O(\eta^2), which is dominated by \Sigma_post \in O(\eta)?

---

> ### Author Response · Authors · 2018-11-15
> **Response (1/2)**
>
> First, it should be noted that neither Khan et al. (2018) nor Zhang et al. (2017) recover the root-mean-square-gradient form for the normalizer.  To quote from Khan et al. (2018):
> "Using ... an additional modification in the VON update, we can make the VON update very similar to RMSprop. Our modification involves taking the square-root over s_{t+1} in (7)"
> And to quote from Zhang et al. (2017):
> "These update rules are similar in spirit to methods such as Adam, but with the addition of adaptive weight noise. We note that these update rules also differ from Adam in some details: (1) Adam keeps exponential moving averages of the gradients, which is equivalent to momentum, and (2) Adam applies the square root to the entries of f in the denominator. We define noisy Adam by adding momentum term to be consistent with Adam. We regard difference (2) inessential. The choice of squaring or divison may affect optimization performance, but they don't change the fixed points, i.e. they are fitting the same functional form of the variational posterior using the same variational objective."
> Both of these approches use natural gradient VI, and they both encounter the same problem: that natural gradient gives you a mean-square-gradient, rather than a root-mean-square gradient form for the normalizer.  Khan et al. (2018) deal with this by reaching in and replacing the mean-square with a root-mean-square normalizer (without a principled justification based on approximate inference), and Zhang et al. (2017) regard the difference between a root-mean-square and a mean-square gradient normalizer as "inessential".
> Moreover, neither approach gives rise to momentum.
>
> Olliver (2017) works hard to ensure that their filtering technique is equivalent to natural gradient VI, and therefore, while they don't examine it, they are also likely to be unable to recover the root-mean-square normalizer.  More problematic is the (as you quoted) "the addition of an artificial process noise".  However, it is important to complete the quote, "the addition of an artificial process noise Q_t proportional to [the posterior covariance] P_{t-1}"  There are two things that they could potentially be referring to as "artificial" here:
> 1.) The introduction of any process noise into the generative process.
> 2.) The introduction of a generative process that depends on inferences under that process.
> Notably, 2 is very, very artificial: I have never seen a Bayesian generative model which depends on inferences under that model.  Our approach does not use this process noise, and as such, if nothing else, our generative process is more meaningful, in the very basic sense that it doesn't depend on the posterior.
>
> Detailed comments:
>
> (1) There is no such equivalence, because, as we make clear in a new section, entitled "Factorisation implies a rich dynamical prior", these two methods don't even perform inference over the same random variable. The approach that you suggest towards factorising the problem fails immediately because P(w_i| D) (i.e. the distribution over one parameter conditioned on all the data), is meaninglessly broad distribution, because the posterior distribution is highly symmetric (e.g. unit-swapping symmetries).  Further, under this model, there is no principled motivation for the introduction of dynamics.
>
> Instead, we note that the updates in most algorithms for neural network optimization are based on just the gradient for that parameter.  As such, if we define a factorised model for each parameter separately, we can take the "data" to be not the underlying input-output pairs, but the backpropagated gradient for that parameter.  This generative model naturally gives rise to the emergence of a new latent variable, w_i^*, the optimal value for the ith parameter, conditioned on the current estimate of all the other parameters.  If we choose to work with this parameter, then dynamics emerge automatically: w_i^* must change over time, because it depends on our current estimate of all the other parameters, which are changing as they are optimized.
>
> In essence, this approach converts the intractable high-dimensional correlations in the full posterior into tractable low-dimensional temporal dynamics.  Notably, dynamics do not and cannot emerge from a straightforward factorised approximation to the original full posterior.  I'd be grateful for any further assistance to clarify the presentation.
>
> Minor: We have rewritten this section, pushing the discussion of Fisher Information into the Appendix, and defining e e^T such that it equals \mLambda.  This is possible because the Hessian is always rank 1 (as we now clarify).

---

### Official Review · AnonReviewer1 · 2018-11-02
**Promising novel research, high practical relevance**

**Rating:** 7
**Confidence:** 4

**Review:**

* Description

The paper considers the following random process on the parameters z (modeled as Gaussians):
- shrink z towards zero and add Gaussian i.i.d. noise to it.
- update the parameters to the posterior w.r.t. a batch, where the likelihood is approximated as a diagonal multivariate normal distribution.
This results in a Kalman filter like updates. There have been related methods proposed performing Bayesian learning in the form of assumed density filtering, considered as separate learning algorithms. At the same time methods such as RMSprop and Adam were previously derived from completely different considerations. The work can derive these methods in the Bayesian framework with certain additional assumptions / simplifications. It allows to naturally explain tracking the gradient statistics as uncertainties and the normalization of the gradient in the existing methods as the update of the mean parameters in the Kalman filter taking into account these uncertainties.
The experiments on MNIST show that derived more Bayesian variants of RMSprop and Adam can improve generalization in terms of test likelihood and test error.

* Assessment

The provided derivation of Bayes like learning algorithms is relatively simple and could be very useful in practice and in further improvement of the learning methods. The approximations used are not completely clear. The clarification of the idea of a separate optimization problem per variable is necessary. The provided experiments, if there is nothing subtle, are clearly done and would be sufficient.
There are some open questions such as: does the method in fact learn useful variances of the parameters, i.e. really performs an approximate Bayesian learning? Overall if find it a promising novel research direction of high practical relevance.

* Clarity

Intro:
Why is the unnumbered equation on page 1 is called a “Bayesian optimization problem”? There is so many sings called Bayesian that one cannot be sure what it means. In the context of the paper it should be a Bayesian learning problem, but I do not see a posterior distribution over the parameters. Overall, I did not get the point of the discussion in the introduction and Figure 1 altogether. Everything it says to me is that global minimize coordinates are dependent through the objective. I do not see what the unnumbered equation on page 1 has to do with Bayesian inference and how the correlation of parameters in the posterior distribution is related to the dependencies in the minimizer. Could authors please seriously consider clarifying this section?
In what follows the paper keeps a factorize approximation to the posterior of parameters of a NN in the form of a Gaussian distribution per coordinate. It thus does not in any way avoid making this restrictive assumption.

Results:
Sorry, I am not familiar with the background behind (6). Which value of z is assumed in the conditional expectation, is it conditioning on “z = \mu_{prior}”? How come the approximation to the variance of the data likelihood does not depend on the data? If we make this approximation, how much it is still relevant to the Bayesian learning?

What are the overheads of the proposed methods? I expect they scale as easily to large problems as SGD?

* Experiments

From Figure 2 it seems that BRMSprop and BAdam can achieve relatively good results for large range of eta in 10^-5 to 10^-2 and it seems from the trend that even smaller eta would work. Does it mean they do not need in fact tuning of the learning rate?
The experiment uses 50 epochs, do the compared methods reach the convergence? Could the authors consider an experiment running best setting of parameters per method with twice as many epochs?
Some artificial toy experiments could be of interest. For example, consider a classification problem with a 1D Gaussian data distribution in each class and the logistic regression model with 2 parameters. Does the method approximate the posterior distribution?

* Related work

The approach to Bayesian learning taken in the paper needs to be better discussed. I think it is from the family of methods known as “assumed density filtering”, occurring in:
Ghosh et al. “Assumed Density Filtering Methods for Scalable Learning of Bayesian Neural Networks”
with earlier works well described in
Minka T. “Expectation propagation for approximate Bayesian inference”.
In particular equation (5) of the submission is well known.
The work  Khan et al. 2018 “Fast and scalable Bayesian deep learning by weight-perturbation in Adam” also derives Bayesian learning algorithms in the forms closely similar to RMSprop and Adam and interprets the running statistics as uncertainties. However it takes the variational Bayesian learning approach, which means the reverse KL divergence is used somewhere. Could the authors discuss conceptual similarities and differences to this work?

---

> ### Author Response · Authors · 2018-11-15
> **Response (2/2)**
>
> Experiments: usually, when we set the learning rate to be too small, it is impossible to move the parameters far enough from their initializations, and so we obtain very poor performance.  One of the interesting things about our framework is that as eta goes to zero, it converges to Bayesian inference without the dynamical prior, which, in effect, gives an adaptive learning rate that goes as the sum-of-square gradients.  In practice, we still expect to need to tune the learning rate to find the minimum of the curves in Fig. 2 (now Fig. 3).  As regards toy experiments, we expect the benefits of our approach to become more evident in complex models with strong posterior correlations, so the simplest relevant toy experiment is linear regression with highly correlated inputs --- which isn't all that simple.  Regarding convergence, tried a range of numbers of epochs, and found that the displayed curves are pretty stable at 50 epochs, but that if anything, the difference between the Bayesian and classical methods actually increased for larger number of epochs.
>
> We have discussed the ADF-style approach (Ghosh et al.), and have noted that the Kalman filtering approach is well-understood. Finally, it should be noted that Khan et al. (2018) does not recover the root-mean-square-gradient form for the normalizer.  To quote from Khan et al. (2018):
> "Using ... an additional modification in the VON update, we can make the VON update very similar to RMSprop. Our modification involves taking the square-root over s_{t+1} in (7)"
> Both they and Zhang et al. (2017) of these approches use natural gradient VI, and they both encounter the same problem: that natural gradient gives you a mean-square-gradient, rather than a root-mean-square gradient form for the normalizer.  Khan et al. (2018) deal with this by reaching in and replacing the mean-square with a root-mean-square normalizer (without a principled justification based on approximate inference), and Zhang et al. (2017) regard the difference between a root-mean-square and a mean-square gradient normalizer as "inessential".

---

> > ### Comment · AnonReviewer1 · 2018-11-19
> > **Further clarification requests**
> >
> > I appreciate very much what authors explained in their response and the improvements made to the paper. The following details need further clarification in my opinion.
> >
> > I understand the the issue of symmetries in model parameterization is avoided by considering the Laplace approximation to the posterior (a multivariate Gaussian around a mode of the likelihood). Furthermore, because the factorization of the posterior is an undesirable assumption, it is proposed to consider the distribution (1) and its Laplace approximation, which matches the respective conditional distribution of the global Laplace approximation.
> >
> > The mode of this approximation is very unfortunately denoted the same way as the random variable itself, w*_i. Why not to use mu_i for the conditional mode? Also, expanding the log likelihood as the expectation seems unnecessary at this point.
> > The following notation is confusing:
> >
> > P (w*_i | w_{−i} = µ_{−i}, D)
> > because w*_i is already defined as a shorthand for w_i | w_{−i} = µ_{−i}, D in (1). The conditioning accomplishes nothing.
> > The argument of this distribution being intractable  is unclear because it is a 1D distribution and given the data its empirical distribution or moments can be computed.
> >
> > P(w*_i | D) - this does not make sense to me if definition (1) is used. If the authors want to speak about the marginal distribution P(w_i | D), it has been already discussed in the beginning of sec. 2.
> >
> > Equation (3) models the predictive probability for each batch of data as Gaussian. This would break down for e.g. batch size 1 and logistic model. I.e. some comment on batch size is needed. After this assumption is made, the distribution of  the whole data D is seen to be Gaussian (as a product of Gaussians over all batches) as well as the posterior distribution w_i | w_{-i}, D as desired, these distributions should be possibly given instead of the chain (6)-(7). In particular this will define w^*_i and Lambda_i as parameters of
> > w_i | w_{-i}, D under this approximation.
> >
> > In (6), I do not understand µ_{like,i} | Lambda_i, w^*_i. Does this mean conditioning on the r.v. w*_i being equal to the mode of its approximate distribution w*_i? This is very confusing. The remark that "w*_i must be a valid sample" is not helpful at all. It is further unclear which distribution is assumed and which is being approximated. The right hand side Lambda_i cannot be the batch-dependent Lambda_i in (4), can it?
> >
> > The discussion about factorization matter reduces in the end to the difference in estimating
> > the marginal distribution of the Laplace approximation of w | D (which is a single Gaussian around a mode)
> > versus conditional distribution w_i| w_{-i}, D of that very same approximation at w_{-i} fixed to its mean. Interestingly, the conditional distribution will have the same mean, but different variance from the marginal distribution (and seems to be the one that can be connected to the variance of the gradient).
> > Now with this difference visible I do buy the arguments about coupling of parameters and not making the factorization assumption. However, speaking of multiple independent estimation problems still seems a suboptimal explanation. Can it be replaced with iteratively estimating the conditional distributions?
> >
> > Please address these issues, I believe the intro could be still significantly clarified and made more accessible.

---

> > > ### Author Response · Authors · 2018-11-20
> > > **Response**
> > >
> > > Thanks for your comments.
> > >
> > > Eq (1) and Eq (2) are actually equivalent: I agree, it would be crazy to use w*_i for two different things.  I think of Eq (1) as the "primary" definition of w*_i.  The confusion comes in with Eq (2).  In particular, Eq. (2) is *not* the mode of the usual likelihood, conditioned on all the data.  Instead, the expectation is taken over data, d, drawn from the underlying, unknown, true data distribution (which is why we need to write it as an expectation).  As such, Eq. (2) in effect does maximum likelihood with infinite data, which picks out the true value for the parameter, w*_i.  I have altered the text to clarify this point.
> > >
> > > There was a slight problem with the notation here: we have changed P(w*_i | w_{−i}=µ_{−i}, D) to P(w*_i | µ_{−i}, D).  That said, w*_i as defined by Eq. (1) is a standard random variable, so it can be marginalised and conditioned, just like any other.  In particular, we can marginalise over µ_{−i}, using,
> > > P(w*_i | D) = \int dµ_{−i} P(w*_i | µ_{−i}, D) P(µ_{−i}| D),
> > > while it is difficult to write down the distribution over µ_{−i}, it could be sampled. In fact, this is just what we do in transitioning from Fig. 1A to Fig. 1B: the integral over µ_{−i} is difficult, so instead we directly write down simplified dynamics over w*_i.
> > >
> > > Computing one of these distributions is order(N), where N is the number of parameters, so it should be tractable.  However, computing all the distributions is order(N^2), which is intractable for all but the smallest networks.
> > >
> > > Note that while Eq. (3) takes a "Gaussian-like" form, d doesn't appear in the correct place on the RHS.  Instead, this is a Gaussian over a parameter, w_i.  What's going on here is that we're only interested in the dependence on w_i, so we have put all the dependence on the minibatch into const, mu_{like,i} and Lambda_i.  As these are arbitrary functions of the data, we can, in fact have arbitrary data.  More broadly here, we're treating the likelihood as function of the parameters (e.g. http://www.inference.org.uk/mackay/Bayes_FAQ.html#likelihood).
> > >
> > > mu_{like,i} is the mode for one minibatch.  w*_i is expected mode, given infinite data.  Because mu_{like,i} depends on the minibatch, it will exhibit some variability across minibatches, mu_{like, i}, but it will have mean w*_i.  I have updated this section, to note that we can also get the variability of mu_{like, i} by considering the Fisher Information, which gives us the variance of g.  Lambda_i can be batch-dependent (though the derivation simplifies considerably, if we assume that it is fixed, and I did this initially).  For instance, consider classification with noisy labels (e.g. from mechanical turk), where some labels are known to be high-quality, and others are low-quality.  The model should use higher Lambda_i for high-quality, and thus more informative labels.
> > >
> > > I think the same basic approach could be used in many domains: importing some aspects of the approximations into the inference problem, such that we can use Bayes theorem to reason under the approximations.  However, I'm not sure this would simplify the presentation much.

---

> > > > ### Comment · AnonReviewer1 · 2018-11-20
> > > > **Clarity / Technical correctness**
> > > >
> > > > I thank the authors for their response. I will consider it in more detail when I have time. At the moment the following is important. I like very much the idea of the paper. However, when it comes to the final recommendation, clarity and technical correctness are necessary. At the moment the paper says:
> > > >
> > > > w*_i ~ P(w_i | ...),
> > > > which I read as "w*_i is a r.v. distributed as P(...)". Right next, it is
> > > > "Note that w*_i  is close to, but not quite the mode of the likelihood of the actual data"
> > > > However, it cannot possibly be the mode, because it is a random variable. If it was not a random variable, one cannot write P(w∗i|µ−i, D).
> > > >
> > > > At present, due to non-standard and inconsistent notation, it is not possible to follow the work. I mean also from the point of view of the future readers.
> > > >
> > > > I would recommend the following:
> > > > -clearly distinguish random variables and their possible values
> > > > -avoid condition on random variables, only on events (or expect the result of e.g. E[stuff|w*_i] to be a random variable).
> > > > -make all events you condition on clear, for example, conditioning on \mu_i is not clear.
> > > > - check that all entities are defined before they are used. For example, Lambda_i is not defined other than a data-dependent r.v. (4). Nevertheless it is present on the RHS of (9) while not being conditioned on in the LHS. This step in particular, silently omiting Lambda_i in P(g|w*) is currently a gap in the derivation. Making everything precise should help to identify and resolve such gaps.
> > > > - make all claims step-by step verifiable. For example the variance in (6) is not. One have to guess about the meaning, assumptions and the proof.
> > > >
> > > > Also, cannot the expression (6) for the precision of the batch likelihood be used instead of the estimation in (9.7)?
> > > >
> > > > Please take you time to make the work solid.

---

> > > > > ### Author Response · Authors · 2018-11-21
> > > > > **Response**
> > > > >
> > > > > Thanks for your comments.  I have updated Section 2, which hopefully clarifies it considerably.
> > > > >
> > > > > It is important to note that w*_i is both a mode, AND a random variable, because it depends on the current estimate of the other parameters, mu_{-i}, which we treat as a random variable.  I updated the text to emphasise this point, including by deleting Eq. (1), leaving only the mode-based definition.
> > > > >
> > > > > - We have done this as far as possible (e.g. by using Lambda_i = lambda_i).
> > > > > - We have rewritten the text to avoid conditioning on random variables, except where absolutely necessary (i.e. Eq. 1), and for Eq. (1), we have repeatedly emphasised the consequences of conditioning on a random variable.
> > > > > - We have removed \mu_i from the conditioning.
> > > > > - I have given more specific definitions of mu_{like, i} and Lambda_i as Taylor series coefficients, simultaneously with the original Taylor series expansion.  It is difficult to any more specific definitions of mu_{like, i} and Lambda_i without assuming a specific form for the data, which, as you pointed out earlier, is highly undesirable.  I have ensured that Lambda_i is included in the conditioning in Eq. (7)/Eq. (8).
> > > > > - I have removed the discussion of the variance in Eq. (6), and instead noted that typical approximations of the original likelihood (new Eq. 2) are necessarily quite severe (e.g. using the Fisher Information).  As such, instead of trying to obtain new approximations, we instead, try to leverage these older approximations, by obtaining a model for the gradients which is equivalent to any given approximation to the original likelihood (new Eq. 2).  In particular, our goal is that the likelihood, conditioning on the gradient, should be equivalent to whatever approximate second-order Taylor series approximation to the original likelihood in new Eq. (2) one chooses to make.
> > > > >
> > > > > It would be interesting to investigate whether this derivation could be used improve or better understand typical approximations for Lambda_i (or the variance of the gradient).  However, in practice I suspect you are forced to use quite severe approximations, that are difficult to justify without being able to refer to past work that has used those exact approximations, and as such, it is easier to try to use approximations from past work directly.

---

> > > > ### Comment · AnonReviewer1 · 2018-11-21
> > > > **Detailed answers**
> > > >
> > > > > we have changed P(w*_i | w_{−i}=µ_{−i}, D) to P(w*_i | µ_{−i}, D).
> > > >
> > > > Here is how I am trying to parse it.
> > > >
> > > > When you define w*_i by (1), it can mean either
> > > > 1) the shorthand w*_i = (w_i | w_{−i}=µ_{−i}, D)
> > > > or
> > > > 2) w*_i is an independent r.v. that has the same distribution as P(w_i | w_{−i}=µ_{−i}, D)
> > > >
> > > > In case 1), further conditioning of w*_i on D is equivalent to (w_i | w_{−i}=µ_{−i}, D,D), i.e. conditioning on D twice, which is redundant.
> > > >
> > > > In case 2), w*_i is independent of everything else and conditioning is identity.
> > > >
> > > > Conditioning on µ_{−i} can be understood as conditioning on the event w_{−i}=µ_{−i}, but you claim now it is not and mention the distribution of µ_{−i}. Previously it was a current estimate (i.e. a possible value) of w_{−i}. If it is not, I need to start over and ask for the definition of µ_{−i}.
> > > >
> > > > > Note that while Eq. (3) takes a "Gaussian-like" form
> > > > There is no contradiction with my objection. For the logistic classification model and one training example, the likelihood (the probability viewed as a function of parameter) is also logistic. One needs to multiply several of those for training examples from different classes to get something bell-shaped.
> > > >
> > > > > Lambda_i
> > > > Lambda_i is defined as the precision of the Gaussian approximation to the likelihood of one batch in one parameter. You are using other interpretations of Lambda_i, but they are not defined in the paper. " if we assume that it is fixed, and I did this initially" -- fixed to what?
> > > >
> > > > Hope this helps

---

> > > > > ### Author Response · Authors · 2018-11-21
> > > > > **Response**
> > > > >
> > > > > Thanks again for your comments.
> > > > >
> > > > > I updated Section 2 in response to your previous comment, so some of these points may be redundant.  Nonetheless, there are still some important points to be considered here.
> > > > >
> > > > > This has now gone (in response to your next comment), and we define w*_i purely using the argmax.  Please let me know if I can clarify further.
> > > > >
> > > > > We no longer condition on the value of µ_{−i} anywhere.  I agree that doing so would lead to very confusing notation.  That said, µ_{−i} is still random variable controlled by the optimisation of all the other parameters.  It is not possible to write down this process explicitly, because not only are the dynamics of µ_{−i} highly complex, but they also depend strongly on e.g. the optimiser.  We deal with this in the transition from Fig. 1A to Fig. 1B, by writing down a simplified model over w*_i directly, rather than trying to integrate over the µ_{−i} random variables.
> > > > >
> > > > > Agreed about the Gaussian-like form.  However, this is the approximation taken in past work (e.g. Zhang 2017; Khan 2017; 2018), and our goal here is not to improve upon these approximations, but to place them in a dynamical framework.  I have attempted to draw closer links between this past work and the current derivation, but please let me know if I can make these stronger.
> > > > >
> > > > > The comment about Lambda_i has gone.  Broadly, as we are currently trying to tie the derivation in Sec. 2 to past approximations (e.g. Zhang 2017; Khan 2017; 2018), we will choose Lambda_i based on those approximations (e.g. using the Fisher Information).

---

> ### Author Response · Authors · 2018-11-15
> **Response (1/2)**
>
> Assessment: We have hopefully clarified the approximations (esp. in Adam) and the introduction of a separate inference problem per variable (see the new sectinon "Factorisation implies a rich dynamical prior".  It should be noted that the methods we propose --- BRMSprop and BAdam --- don't use these approximations.  Instead, the approximations are only necessary to understand the similarities and differences betweeen BRMSprop and BAdam and the corresponding classical methods: RMSprop and Adam.  Finally, I agree that assessing the effectiveness of the Bayesian model for the optimal weight would make interesting future work, and we envisage including it in a broader investigation of the statistics of optimal weights under optimization.
>
> Intro: we have deleted the reference to Bayesian optimization, and rewritten this section in order to clarify it considerably (see the section "Factorisation implies a rich dynamical prior").  In this section, we note that we aren't making a factorised approximation in the usual sense, where you write down an inference problem over all N parameters jointly, and use a factorised posterior.  Instead, we change the inference problem itself, by writing down N inference problems, one for each parameter, where the data for each individual problem is the backpropagated gradient.  This bakes factorisation into the problem setting.
>
> When we look at the generative process for the gradients, we find that the potential for a rich dynamical prior has emerged automatically, because the gradients for one parameter depend on all the others, and those other parameters change slowly as they are also being optimized. Critically, this converts intractable high-dimensional correlations in the original problem into tractable low-dimensional dynamics.  In contrast, these dynamics do not emerge in a traditional approach where you simply approximate the high-dimensional posterior.
>
> Results: this is an interesting point, which bears further discussion.  I agree that the obvious approach here would be to use the Hessian (i.e. the second derivative of the actual likelihood).  However, using Hessian directly has two problems.  First, the Hessian isn't necessarily positive definite, and if such a likelihood is combined with a sufficiently weak prior under our approximations, it can result in a meaningless posterior (e.g. a Gaussian with a non-positive semidefinite covariance matrix).  Second, the Hessian (i.e. the full matrix of second derivatives) is difficult and expensive to compute in modern autodiff software such as PyTorch.  Because of these two disadvantages, we chose to work with a closely related quantity: the Fisher Information.  This is the Hessian for the expected log-likelihood of data drawn from the model, and it therefore represents an approximation to the actual Hessian.  However, in the case of classification, we expect the approximation to be reasonable, because it is based on the same input data, with output labels sampled from the model.  And as the classification error gets smaller, the model outputs and the true classification become more similar, and thus the approximation becomes becomes increasingly good.  Importantly, the Fisher Information resolves the two issues we had above: it is always positive definite, and it can be computed by taking the covariance of gradients, which are easy to compute in standard autodiff frameworks.  This is a standard approach (e.g. Zhang et al. 2017).

---

### Official Review · AnonReviewer2 · 2018-11-02
**I do not find the results of the paper particularly convincing though I would not rule out Bayesian filtering as a framework for analyzing adaptive methods**

**Rating:** 5
**Confidence:** 3

**Review:**

Paper summary: The authors analyze stochastic gradient descent through the lens of Bayesian filtering. In doing so they (approximately) recover several common adaptive gradient optimization schemes. The paper focuses on a theoretical construction of this framework and offers a limited empirical study.

Detailed comments:

I thought that the paper presented some interesting ideas but amongst the many things discussed there is very little which is empirically gratified. While the Bayesian filtering framework is interesting in that it recovers slight variations of existing algorithms, and also caters for some recent practical tricks, I do not feel that it substantially improves our theoretical understanding of these methods.

1) I found the notation difficult to follow in the introduction and parts of section 2. I have highlighted several places explicitly below. I found paragraphs 2 and 3 of the introduction particularly challenging.

2) I found the introduction of Bayesian filtering challenging to follow. For example, which form of the likelihood is assumed for the Taylor expansion? How/why is $\mu_{like}$ identified using the gradient? Linking to Kalman filtering made things easier to follow.

3) I think that the related work, and possibly a chunk of section 2, should include a discussion of Noisy Natural Gradient [1]. While the derivation differs, the motivation and final form of the updates seem to have a large overlap but this work is not cited.

4) Start of 2.1: "z will have on element representing a single parameter", after which z is treated as a vector. I believe this sentence is present to distinguish RMSProp from Adam when momentum is added but I found it confusing at first.

5) I found the comparisons between BRMSProp-vs-RMSProp and BAdam-vs-Adam fairly unconvincing. The assumptions are not clearly demonstrated to have little practical significance and Figure 2. does not seem to support the claim that these methods are strongly related. Is it possible to demonstrate empirically that these algorithms have equivalent behaviour under some limiting factors? And if not, is there a good reason for this that still justifies the comparison? I would appreciate some clarifications on these points.

6) I am not sure what you mean by "We now assume that the data is strong enough to reduce the uncertainty in the momentum below its levels under the prior". I believe that I am following the mathematical arguments correctly but I find this phrasing misleading. Furthermore, this section uses e.g. ppth and wpth to refer to coordinates, I think it would be clearer to simply write Sigma_{pp}, etc.

7) Section 4.2 is lacking justification in my opinion (am I missing something?). I think that this section needs to have the derivation clearly laid out (in the appendix would be fine). Furthermore, the NAWD algorithm is not explored empirically, or analyzed theoretically at all. I would argue that more evidence is needed that this is a reasonable thing to do before it is meaningful to include it in the final print of this paper. In general, sections 4.4 - 4.7 feel a little out-of-place and thrown together. I think there are interesting comments here which are certainly worth including but their presentation should be rethought and some empirical investigation would be valuable.

Minor comments:

- In introduction, how exactly does $w'_i$ differ from $w_i$?
- In introduction, after para 2, the notation in the equation is confusing, e.g. overloading w_i(t) and w_i(mu_{-i}(t)).
- In introduction,  para 3, "must depend on other parameters" - this seems like an obvious statement but it is presented as being crucial
- Should "Related Work" start at 1 or 2?
- (VERY MINOR) In section 2.2 and 2.3, "christen" seems like an add choice of word. Perhaps just "call"?
- Equations 10 and 11 introduce an independence assumption on the dimensions of the parameter vector. I think this should be explicitly stated.
- Section 7.2 heading typo: MOMEMTUM

Clarity: I found the paper challenging to follow in places due to choices of notation (and a weak background in Kalman filtering and related techniques).

Significance: I do not feel that this work offers a strong case for significance. The empirical evaluation is very limited. The theoretical framework introduces is interesting but is not justified particularly well in the paper and does not directly offer explanations for many of the observations noted in this paper and elsewhere.

Originality: To my knowledge, the ideas presented in the paper are original and hint at potentially interesting viewpoints of optimization.

References:

[1] Zhang et al. "Noisy Natural Gradient as Variational Inference" https://arxiv.org/pdf/1712.02390.pdf

---

> ### Author Response · Authors · 2018-11-15
> **Response (2/2)**
>
> 7) We have added an Appendix section spelling out how inference in this filtering model implements NAG.  The empirical evaluations in Fig. 2 include NAWD as part of the full BRMSprop and BAdam algorithms.  We agree that it would be great to do an empirical investigation of all the features discussed in this paper, but this becomes an empirical analysis of a pretty large swathe of techniques for adaptive stochastic gradient descent, which is out of scope for the present paper.
>
> Minor comments:
> - w' has been removed.
> - While we have rewritten this section, we have retained the notation.  w is the underlying unknown random variable, whereas mu is our current mean estimate of that variable. Hopefully this distinction is clearer in the new version.  We are also using the recommended DL book notation for random variables.
> - I agree, it is obvious.  But it is also crucial: it is the key for why we need to introduce a rich dynamical prior.  Again, hopefully this discussion has been improved in the new version.
> - Related work has been incorporated into the introduction.  I've also explicitly introduced an Introduction section, so that next section is 2.
> - I have replaced christen
> - The independence assumption comes right at the start when we write down a factorised generative model (as we explain in the new section "Factorisation implies a rich dynamical prior"  in Eq. 10 in the original submission, we're doing filtering in a 1D inference problem.
> - Fixed

---

> > ### Comment · AnonReviewer2 · 2018-11-20
> > **Thank you for the response. I acknowledge improvements but key concerns remain for me**
> >
> > I think the introduction is much better and, to my knowledge, the work is now much better placed among existing work.
> >
> > 1) I think Section 2 could make a good addition but in its current form is poorly written and difficult to follow.
> >
> > Notation in section 2 is confusing (as pointed out by Reviewer 1). In particular, w*_i seems very overloaded. Is it a sample from the posterior or a maximum likelihood estimate or does it denote a random variable?
> >
> > What are unit-swapping symmetries exactly? Please explain or add a citation if relevant.
> >
> > (minor) "Therefore, we can consider we need to consider..."
> >
> > Eqn 2: Little `d` instead of `mathcal{D}`? Or do you mean the minibatch notation added later?
> >
> > "Remember that the updates for almost all neural network optimization algorithms depend only on the backpropagated gradient for that parameter." What does this mean? When using momentum we depend on the history. Regularization also affects this, e.g. weight decay.
> >
> > 4) I still think referring to vectors as elements is confusing, but Equation 10 helps.
> >
> > 5) Thank you for clarifying these points. I now better understand the motivation for the empirical evaluation but I still hold some reservations. If the aim is to show that using well motivated updates we can achieve improved optimization then a much more thorough empirical evaluation is required (or perhaps a stronger theoretical argument for this point). I do not consider the MNIST classification results to be a strong indicator of the optimization performance of these algorithms.
> >
> > 6) Thank you for clarifying. I now understand the assumption at a high level.
> >
> > 7) I think that this point should be clarified in the main text. I see that you do not claim to directly recover NAG but this was how I initially read this section. You comment that the updates adopt a very similar form to Nesterov accelerated gradient, with the addition of root-mean-square normalization. But this is a critical difference and one that I don't think can be ignored. I acknowledge that this prior gradient evaluation is indeed evaluated in Figure 2.
> >
> > I can understand your point with regards to my expectations for empirical evaluation. You do introduce many new techniques and it would be unreasonable to expect you to comprehensively investigate every single one of them. However, I do believe that some more evidence is needed to support some claims in the paper. There are many approximations made throughout and while these may be standard you are applying them in a novel setting. I think that some empirical (or theoretical analysis) is due to justify these. Several mechanisms are identified and related to existing work but deep learning optimization is subtle and it is not yet obvious that these are responsible for any empirical successes of BAdam/BRMSProp. I stand behind my previous comment that (the now) Section 7 introduces many (interesting) observations but without any empirical evaluation these contribute little to the paper.
> >
> > To summarize, my opinion has been largely unchanged. There have been many improvements to the clarity of the paper but unfortunately I think that the new Section 2 undoes this in part. The theoretical framework is certainly interesting but I believe some effort should be taken to evaluate these optimizers on more challenging tasks than MNIST classification. Or even better, experiments should be designed to understand how the qualities of these algorithms help in practice.
> >
> > I would be pleased to partake in additional discussion.

---

> > > ### Author Response · Authors · 2018-11-21
> > > **Response**
> > >
> > > Thanks for your comments.
> > >
> > > (1) I have updated section 2 considerably, also in response to comments from AnonReviewer1.  We now define w*_i only through the argmax.  However, w*_i is still a random variable, because it depends on on the current settings of the other weights, mu_{-I}, which are treated as random variables.  Please let me know if this needs further clarification.
> > >
> > > Unit-swapping symmetries is where we permute the hidden units, and the weights and biases such that the output of the network is unchanged.  For instance, see Sussmann (1992) "Uniqueness of the weights for minimal feedforward nets with a given input-output map."
> > >
> > > \mathcal{D} referred to the actual data.  Here, we mean to use d, which represents a single minibatch, randomly chosen from the true underlying data distribution.
> > >
> > > (5) Many state-of-the-art methods (RMSprop, Adam, AdamW) were initially derived without a strong theoretical motivation.  The primary contribution of this work paper is to indicate that it may be possible to give a principled motivation for these methods.  As such, perhaps we're already seeing good performance in e.g. Adam because Adam has a principled basis.  As such, the empirical evaluations are intended merely to raise the possibility that we might get further performance improvements by explicitly considering this principled approach.
> > >
> > > (7) I have clarified this point in Section 7.
> > >
> > > In Section 7, we draw links to previously suggested, empirically successful methods.  Given that the original papers have already shown the empirical effectiveness of these methods, it is unclear what our simulations might add, over and above this original work.  In order to maintain this strong link to past empirical work, I have deleted the more speculative parts of this section, 7.5-7.7.
> > >
> > > That said, I agree that it would be great to try more challenging tasks: and I'll see whether I have a chance to do this before the deadline.

---

> > > > ### Comment · AnonReviewer2 · 2018-11-27
> > > > **Updated review**
> > > >
> > > > I am sorry for responding late -- I wrote a comment but I guess I did not upload it correctly.
> > > >
> > > > Thank you for clarifying some of these points. I felt that the later revisions and your comments helped to clear up some of the issues I had and have thus updated my review score from 4 to 5 (actually, I did this a few days ago). While I would not be disappointed if the paper were accepted, I still believe that some of my existing concerns hold.
> > > >
> > > > I am optimistic about the future of this work. It is exciting to reconcile momentum in adaptive methods with a principled framework for analyzing these optimization schemes. While I am not entirely convinced currently I would be excited to see how the Bayesian counterparts to the existing momentum schemes compare under a more careful empirical analysis.
> > > >
> > > > As a final minor comment, I think that the more speculative parts could have been moved to the appendix instead of being deleted. I never meant to imply that these were not interesting but rather were not well supported in their previous position.

---

> ### Author Response · Authors · 2018-11-15
> **Response (1/2)**
>
> To my knowledge, this is the first work that is able to reconcile Bayesian inference with adaptive SGD methods such as RMSprop or Adam.  (Please see the response to reviewer 3, or the new introduction for details).  To do this, we introduce an entirely new approach to relating optimization and Bayesian inference, where we define a Bayesian generative model for the backpropagated gradients.  This model implicitly converts intractable high-dimensional correlations in the original posterior into tractable low-dimensional temporal changes.  Empirically, this approach gives very promising initial results.
>
> 1) I have replaced paragraphs 2 and 3 with a section entitled "Factorisation implies a rich dynamical prior", which takes a far more in-depth look at how writing down a factorised generative model forces us to include a rich dynamical prior.  Also see the response to reviewer 3.
>
> 2) I have replaced the title of this section with Bayesian (Kalman) filtering as adaptive SGD to emphasise the link to Kalman filtering.  Hopefully the new introduction has made this clearer.  Otherwise, the approach is relatively standard (e.g. Zhang et al. 2017).
>
> 3) We have included a discussion of [1] in the introduction.  In short, they state: "These update rules are similar in spirit to methods such as Adam, but with the addition of adaptive weight noise. We note that these update rules also differ from Adam in some details: (1) Adam keeps exponential moving averages of the gradients, which is equivalent to momentum, and (2) Adam applies the square root to the entries of f in the denominator. We define noisy Adam by adding momentum term to be consistent with Adam. We regard difference (2) inessential. The choice of squaring or divison may affect optimization performance, but they don't change the fixed points, i.e. they are fitting the same functional form of the variational posterior using the same variational objective."  Thus, they cannot be considered as recovering Adam from natural gradient VI, because natural gradient VI doesn't give momentum or the root-mean-square form for the normalizer.
>
> 4) I've flipped the sentence around so hopefully it's a bit less confusing.  It now reads:
> "For Adam, z will have two elements representing a parameter and the associated momentum, whereas for RMSprop it will have only one element representing a single parameter."
> I have also added a new equation defining z for RMSprop and Adam here, to clarify the shape of z in the subsequent derivation.
>
> 5) I was also surprised to see such substantive performance improvements with BRMSprop and BAdam, as compared with RMSprop and Adam: this is the benefit of obtaining updates as inference under a well-motivated prior.  The justification for the comparison comes from the sections "Recovering RMSprop from BRMSprop" and "Recovering Adam from BAdam".  The correspondence is close for RMSprop (becoming exact as t increases, if g^2 remains constant), and somewhat less so for Adam.  But our goal wasn't to match RMSprop and Adam exactly.  Our goal was to improve upon RMSprop and Adam by capturing their essential dynamical assumptions under a well-motivated Bayesian prior.  As such, there's a sense in which we don't actually want RMSprop and BRMSprop to be exactly equivalent, because if they were equivalent, we wouldn't be able to get inspiration for new adaptive methods, and we couldn't get improved performance.
>
> Otherwise, in Fig. 2, we attempted to make these comparisons as fair as possible otherwise (same network, same initialization, same momentum for Adam and BAdam, sweeping out all sensible learning rates for both methods).  As such, it is also possible to take Fig. 2 (now Fig. 3) as showing considerably improved performance over standard adaptive baseline methods.
>
> 6) In the supplementary section entitled "Setting the momentum decay", we show that under the prior, the uncertainty in momentum is eta_p/2.  Here, we make the assumption that data is reasonably strong, and so our uncertainties are well below those under the prior, i.e. Sigma_pp << eta_p/2.  If this were not true, the network would be unlikely to be able to learn anything.  As such, we might expect BAdam and Adam to differ in the case where the data is weak, and it would be interesting to establish which approach has better empirical performance in that case.
>
> I have replaced references to the ppth element with the "lower-right" element of the matrix equation, and explicitly stated the component equation that I'm referring to.

---

### Comment · AnonReviewer3 · 2018-11-21
**Further clarification requests**

Thanks for the updates. This version is more clear than the previous version.

(1) On page 1, " they modified their natural gradient VI updates, without a principled justification based on approximate inference, to incorporate momentum (Zhang et al., 2017; Khan et al., 2018), and the root-mean-square normalizer (Khan et al., 2017; 2018)."
Is "the principled justification" from the "exact" Bayesian point of view? In [4], the justification to incorporate momentum comes from the optimization view under the variational inference framework. (Please see the heavy ball formulation at Sec E.1-E.2 of [4]).

(2) Eq(5), I think the authors should change the last "=" to "\approx" since you do the first-order Taylor expansion of the gradient function on \mu_like and ignore the higher order terms. Is it correct?

(3) Eq(6) is an approximation.  On page 3, "To compute the expected gradient, we assume that the expected value of \mu_{like,i} is independent of \Simga_i". I think this is a strong assumption, which does not hold in general because "\mu_{like,i}, and the negative Hessian, \Sigma_{i}, are random variables as they are deterministic functions of the minibatch, d. ... "

(4) The difficulty in Eq1-9 comes from the fact that the authors want to perform the "exact" Bayesian filtering on the generative process/model with non-conjugate/non-Gaussian likelihood terms. I think the author should connect the derivation to the Laplace approximation and extended Kalman filter.
From the view of variational inference, it is easier to deal with the non-conjugate terms. (Please see the Kalman filter example Eq 67 at Sec E.2 of [2].) The difficult happens due to the non-conjugate terms in the generative process/model. On the other hand, by choosing the right inference process/model (A.K.A. the variational distribution),  each non-conjugate term can be approximated by a conjugate term expressed by the gradient w.r.t. the variational parameter.  Alternatively, the non-conjugate term can be approximated by an NN (A.K.A. the inference network) in [6]

(5) At Sec 9.1, the authors argue that "to obtain a self-consistent solution, we need, \Sigma_{post} ~ O(\eta)".
 Why does this assumption hold? Why does  the self-consistent solution imply  that \Sigma_{post} ~ O(\eta)?  Does the update (such as Eq 15) meet this assumption?  As t -> \initify, can the authors show that  "\Sigma_{post} ~ O(\eta)"? Can the authors show that empirically this assumption holds in the Bayesian RmsProp/Adam case? If this assumption does not hold,   the proposed update cannot recover the root-mean-square normalization even in the limit case. This point should be addressed.
 Note that Sec 9.1 is critical since the authors use the results in Sec 9.1 to recover the non-Bayesian RmsProp and Adam.

(6) Figure 1.C is an approximation of Figure 1.B at least in the RmsProp case and the Adam case. Is it correct?
As mentioned at http://faculty.washington.edu/ezivot/econ584/notes/statespacemodels.pdf, to recover an autoregressive process of any order, the state space must be augmented. For a full order with infinite time steps as suggested in Figure 1.A, the dimensionality of the augmented state space will be infinite.  Note that in the stochastic gradient setting, the original state space is the parameter space. If Figure 1.C is exactly equivalent to Figure 1.B, the dimensionality of z should be |T| \times dim(w^*), where |T| is the number of time steps and dim(w^*) is the dimensionality of the parameter space. However, in Eq 10,  dim(z_{rmsprop}) = dim(w^*).  It is clear that the authors use the first-order Markov approximation in the RmsProp case, where Figure 1.C is an approximation of Figure 1.B.

References
[2] Khan, Mohammad Emtiyaz, and Wu Lin. "Conjugate-computation variational inference: Converting variational inference in non-conjugate models to inferences in conjugate models." arXiv preprint arXiv:1703.04265 (2017).
[4] Khan, Mohammad Emtiyaz, et al. "Fast and Scalable Bayesian Deep Learning by Weight-Perturbation in Adam" (2018)
[6] Johnson, Matthew, et al. "Composing graphical models with neural networks for structured representations and fast inference." Advances in neural information processing systems. 2016.

---

> ### Author Response · Authors · 2018-11-22
> **Response**
>
> Thanks for your comments.
>
> (1) You’re correct.  To optimize a VI objective, we can use any arbitrary optimiser.  Adam may work really well in practice.  But if we simply use Adam to optimize a variational objective, that doesn’t tell us why Adam might be a good idea in the first place.
>
> There’s a bunch of theory around the qualities of standard natural gradient (with no momentum), and around Bayesian filtering.  Thus, Khan, Zhang and Olliver do give a good justification for the rules that emerge directly from standard natural gradient (or the equivalent Bayesian filtering approach).  In contrast, there is to my knowledge, no theory around “natural momentum” (though it does seem like an important research direction).
>
> (2) We have fixed this, and propagated it through into later expressions.
>
> (3) I have changed this to an approximation, and noted in the text that we get equality when the expectation of mu_{like,i} is independent of Lambda_i.
>
> (4) In general, this is a good idea, as it helps to clarify that the approximations we're making here are those used in past work. In Eq 1-9, we're now explicitly linking to the Laplace approximation / EKF.  In particular, Eq (2) is the Laplace/EKF like-approximation (i.e. second-order Taylor series expansion of the likelihood), and we now note this explicitly.  Further, in Eq. 9 we explicitly link the result back to the original Laplace approximation.  Please let me know if there's anywhere I should further clarify the link.
>
> I should also note that the resulting algorithm is closely linked to these natural-gradient VI style approaches.  As you noted initially, we recover them through Olliver (2017) in the limit as eta -> 0.
>
> (5) Thanks: this is a good point.  It’s only self-consistent with respect to the steady-state expressions, which I wasn’t pointing out here.  I’ve given a considerably more thorough derivation, showing where the scaling comes from, though it becomes a good-bit more involved (e.g. requiring us to make a distinction between big-O and big-Theta, as in Knuth (1976) “Big omicron and big omega and big theta.”).
>
> (6) You're right that, in essence, this can be understood as an approximation.  There's a couple of points to make.  First, an approximation is absolutely necessary here, as we cannot write down and/or integrate over the distribution over all the other parameters, mu_i.  Second, in some ways, I wouldn't go so far as to call it an approximation.  In particular, for an approximation, we should have written down the exact dynamics of mu_i, and simplified them.  Here, we have given up on this exercise entirely, and instead written down a model directly over z.  This is what we mean when we say "In practice, we use a simplified model as reasoning about all possible trajectories of μ_{−i}(t) is intractable".

---

### Comment · AnonReviewer1 · 2018-11-23
**new version -- still some gaps**

The formal readability has improved and the definition (1) is ok now. It could be clarified that parameters \mu are random due to the use of stochastic optimization methods. The comment about "dummy" w_i is unnecessary as it is a bound variable of argmax.

(Q1) "To obtain an informative, narrow posterior..."
With definition (1), the variable w^*_i is only random because mu are assumed random. If one considers a non-stochastic optimization method such as full gradient descent or quasi-Newton, then mu are not random. In this case w*_i has a delta distribution at argmax, and we get no informative posterior. The distribution of w*_i alone thus is not connected to the Bayesian posterior / Laplace approximation but describes some other stochasticity. Even in the case stochastic optimization is used, as soon as we require convergence, parameters mu converge to some point and the distribution of w*_i approaches a delta distribution again. This alone does not lead to an "informative posterior".
It should be clarified that w*_i seeks the mode but we also want to take into account its stochasticity due to other parameters. However, the section (2) only conditions everything on mu_{-i} or on all other the quantities that depend on it, and not quite uses or clearly shows where the stochasticity due to mu-i comes into play. Therefore it is not clear how definition (1) specifying some distribution of w_i* is useful.
What is further confusing, is that in section (3) the hidden state variable is z_i = w*_i has a different meaning from (1). Its distribution is modelled explicitly by its mean and variance. The updates incorporate the data likelihood into the posterior of this variable, including the precision of one batch lambda_i. This is in contradiction with definition (1), i.e. models some other distribution and not (1) as (1) is not related to the posterior of the likelihood. The two views do not quite stitch together.

Some further comments...

"The data depend on the full weight vector, w"
The data is given

(Q2) Expression (6) is very cryptic.
\mu_like depends on Lambda_i and on w*_i only through \mu_-i. The expectation is therefore taken w.r.t. batches d and \mu_-i conditioned on the fixed values of Lambda_i and w*_i. What is happening inside the approximation steps is very unclear. This can be simplified by conditioning directly on \mu_{-i}. Then (6) can be stated as,
E [\mu_like | \mu_{-i}]  \approx w^*_i,
where both \mu_{-i} and w^*_i are r.v. straight font.  Comparing (3) and (1), this approximation means that the expectation of argmax (3) is approximated with the argmax of the expectation (1).
The approximation would be exact if p(d|w) was Gaussian with mean/mode w_i*, in this case all precisions Lambda_i would be the same. I guess this is related to what authors meant by conditioning on Lambda_i and w_i*.
Now, also it follows that
E[\mu_like] = E[w^*_i],
which may be useful, finally connecting to the statistics of w_i*. The expression (7) can be updated respectively as follows. Given \mu_-i Lambda_i and \mu_like are independent,
E[g | \mu_-i] = E[Lambda_i | \mu_-i] (w_i* - \mu),
where booth Lambda_i  and w_i* are r.v.

(Q3)
Lambda_i in (4), to be consistent, should be denoted Lambda_{like,i}. This may resolve some confusion in speaking about precision of the Likelihood of one batch and of the whole training data, which are clearly different thing.

The precision in (8) has been left now mostly unexplained. It is a bit subtle that N(,\lambda) actually denotes normal distribution with variance (and not precision) equal to lambda Maybe saying it in words would be cleaner.
Can it be exactly derived under the Gaussian assumption, ie, consider a Gaussian X with mean \mu and precision \lambda. The precision of the Likelihood of batch of size m (which is readily a Gaussian, the Laplace approximation is exact) is m*lambda and of the likelihood of the whole data of size n is n*lambda and so on.
Under approximation (54) with e = g-\bar g and with the mean "<e^2>" suddenly occurring in RMSprop, the batch likelihood precision lambda_i is de facto being approximated as Var[g]. This is overlapping with specifying the precision of g in (8). If (8) is accepted, the precision is readily Var[g].

---

> ### Author Response · Authors · 2018-11-24
> **Response**
>
> Thanks for your comments.
>
> I have removed the comment about "dummy" w_i
>
> Q1) I have clarified in the text that w*_i is not the optimal weight wrt the data we actually have.  Instead, it is the optimal weight wrt the true underlying, unknown data distribution. As such, this Eq. (1) is analogous to maximum likelihood with infinite data.  Thus, you only get a delta-posterior if you have infinite data.  In contrast, in the usual finite data setting, w*_i is unknown, even if mu_{-i} is fixed, so we obtain a sensible non-delta posterior.
>
> Indeed, the two sections are intended to be quite distinct.  Broadly, Sec. 2 reasons about the generative process for the gradients, whereas Sec. 3 describes how we might perform inference under this model.  In particular, Sec. 2 gives a justification for the dynamical model presented in Eq. (11) (now Eq. 12).  It does this by noting that mu_{-i} introduces not only randomness, but also dynamics, because mu_{-i} changes slowly over time as it is being optimized (Fig. 1A).  However, mu_{-i} is a highly complicated, high-dimensional distribution, we cannot reason about it directly.  Instead, we define a surrogate model (Fig. 1B/C) that captures the structure of the model in 1A, but is highly simplified, such that we can reason about it.  I have changed the boundary between these sections slightly to emphasise that Sec. 3 describes inference in the generative model obtained in Sec. 2.
>
> Further comments:
> I have clarified that we consider the Bayesian setting, where the data (that are indeed given) are assumed to have be generated from the specified generative model.  In this context, maximizing P(data| w) is straightforward, because we have such a model describing the probability of the data in terms of w, whereas maximizing P(data| w*) is much harder, because we don't have such a likelihood.
>
> Q2) I have spelled out a bit more completely the logic here, adding an extra display equation.  While I agree that your approach would simplify matters, in practice, we need to keep conditioning on Lambda_i, because we're trying to follow the usual Laplace approximations to the likelihood, and they use a variety of different approximations for \Lambda_i.  We also need to condition on w*_i, because we're going to end up trying to infer w*_i from the gradient, so we need a model of the form P(g_i| w*_i) (or similar).
>
> Q3) I have moved back to using Lambda_{like,i}.
>
> I've clarified that the gradient is proportional to the negative Hessian, and noted the link to Fisher Information type results in the text.  However, the formal link is a bit technical, and it remains unclear under what conditions this link is formally correct.  The real goal here, though is to be able to use any approximation to the negative Hessian used in past work directly, rather than having to rederive it.

---

### Meta-Review · Area_Chair1 · 2018-12-10
**a promising but heuristic framework for analyzing various optimizers, with little empirical justification**

**Confidence:** 5
**Recommendation:** Reject

**Metareview:**

The aim of this paper is to interpret various optimizers such as RMSprop, Adam, and NAG, as approximate Kalman filtering of the optimal parameters. These algorithms are derived as inference procedures in various dynamical systems. The main empirical result is the algorithms achieve slightly better test accuracy on MNIST compared to an unregularized network trained with Adam or RMSprop.

This was a controversial paper, and each of the reviewers had a significant back-and-forth with the authors. The controversy reflects that this is a pretty interesting and relevant topic: a proper Bayesian framework could provide significant guidance for developing better optimizers and regularizers. Unfortunately, I don't think this paper delivers on its promise of a unifying Bayesian framework for these various methods, and I don't think it's quite ready for publication at ICLR.

There was some controversy about relationships to various recently published papers giving Bayesian interpretations of optimizers. The authors believe the added value of this submission is that it recovers features such as momentum and root-mean-square normalization. This would be a very interesting contribution beyond those works. But R2 and R3 feel like these particular features were derived using fairly ad-hoc assumptions or approximations almost designed to obtain existing algorithms, and from reading the paper I have to say I agree with the reviewers.

There was a lot of back-and-forth about the correctness of various theoretical claims. But overall, my impression is that the theoretical arguments in this paper exceed the bar for a primarily practical/empirical paper, but aren't rigorous enough for the paper to stand purely based on the theoretical contributions.

Unfortunately, the empirical part of the paper is rather lacking. The only experiment reported is on MNIST, and the only result is improved test error. The baseline gets below 99% test accuracy, below the level achieved by the original LeNet, suggesting the baseline may be somehow broken. Simply measuring test error doesn't really get at the benefits of Bayesian approaches, as it doesn't distinguish it from the many other regularizers that have been proposed. Since the proposed method is nearly identical to things like Adam or NAG, I don't see any reason it can't be evaluated on more challenging problems (as reviewers have asked for).

Overall, while I find the ideas promising, I think the paper needs considerable work before it is ready for publication at ICLR.